# On the Theory of Neural Network Surjectivity: Can You Elicit Any Behavior from Your Model?

## Abstract

Given a trained neural network, can any specified output be generated by some input? Equivalently, does the network correspond to a function that is surjective? In generative models, surjectivity implies that any output, including harmful or undesirable content, can in principle be generated by the networks, raising concerns about model safety and jailbreak vulnerabilities. In this paper, we prove that many fundamental building blocks of modern neural architectures, such as networks with pre-layer normalization and linear-attention modules, are almost always surjective. As corollaries, widely used generative frameworks, including GPT-style transformers and diffusion models with deterministic ODE solvers, admit inverse mappings for arbitrary outputs. By studying surjectivity of these modern and commonly used neural architectures, we contribute a formalism that sheds light on their unavoidable vulnerability to a broad class of adversarial attacks.

## 1 Introduction

Deep generative models have achieved remarkable success in recent years—spanning natural language processing (OpenAI et al., 2024; Touvron et al., 2023; Chowdhery et al., 2023), computer vision (Imagen-Team-Google et al., 2024; Grattafiori et al., 2024), and robotics (Kim et al., 2024; Team et al., 2025). Yet this progress has raised growing safety concerns. Powerful models can be manipulated to produce undesirable or even dangerous content (Zou et al., 2023; Wan et al., 2023; Ma et al., 2024a), and the risk only intensifies as their capabilities expand. To mitigate these threats, considerable effort has been devoted to data curation and safety fine-tuning Grattafiori et al. (2024); OpenAI et al. (2024) with the aim of constraining model behavior during training. But a fundamental question remains unanswered: can we ever guarantee that a trained model will not generate harmful content? Or could it be that, given a trained generative model and an arbitrary target output, there always exists an input that produces that output? In mathematical terms, *viewing a generative model as a function from its input space to its output space, we ask whether that function is surjective?*

Formalizing the study of surjectivity in trained models presents significant challenges and deviates from the standard practice of the community. On the one hand, it is clearly too much to expect that *every possible* choice of the parameters yields a model that is surjective. Pathological cases — such as setting all weights to zero — can lead to degenerate models that implement constant functions. But merely observing that some parameter settings lead to surjectivity is equally uninformative, since there is no guarantee that the training process will uncover them. To address this, we adopt a probabilistic perspective: rather than asking whether *all* or *some* parameter settings yield surjectivity, we ask whether surjectivity holds *almost always*. That is, for a fixed model architecture, do all except for a measure-zero subset of parameter configurations, lead to functions that are surjective? This formalism better reflects the practical realities of the training process: the evolution of parameters depends intricately on the choice of optimizer, data distribution, loss function, and even future training paradigms. These elements introduce randomness into the training process, making the final trained model effectively a draw from a high-dimensional distribution. If surjectivity holds almost everywhere, this implies that regardless of the fine-grained details of the training process, it is exceedingly unlikely that the resulting model would not be surjective.

**Technical Results and Toolkit.** Given the significant departure from existing paradigms of studying neural networks, our perspective also calls for a new toolkit to analyze the input-output behavior of trained models. We show that differential topology is the right tool for the job! Differential topology analyzes smooth manifolds under smooth transformations and offers a lens on the global structure of neural network outputs. This connection is far from accidental—modern networks are constructed from smooth components that make them amenable to optimization via backpropagation. As a result, these models are naturally suited to analysis using tools from differential topology. We provide a gentle introduction to some of these tools and show how they can be used to study surjectivity of neural networks with relative ease.

| Architecture | Surjectivity |
|---|---|
| MLP with LeakyReLU | ✔ |
| MLP with ReLU | ✘ |
| MLP with Pre-LayerNorm | ✔ |
| Attention | ✘ |
| Attention with Pre-LayerNorm | ✔ |
| Linear Attention | ✔ |

Table 1: Partial Summary of Results in Section 3. Wrapping a function $f$ in Pre-LayerNorm is defined with residual connection $f(\text{LN}(x)) + x$.

Using these tools, we analyze the surjectivity of core building blocks in modern architectures, including LayerNorm with residual connections, Multi-Layer Perceptrons (MLPs), and Attentions. In particular, we show that any continuous function wrapped with Pre-LayerNorm is surjective (Theorems 3.1 and 3.2)—implying that both Attention and MLP layers are surjective when wrapped in Pre-LayerNorm. We also establish that MLPs with LeakyReLU activation (Theorem 3.3) and linear Attention (Theorem 3.4) are surjective. Two notable exceptions exist: as we show that Attention itself (with soft-max activation) and MLP with ReLU activation are not surjective. We would like to remark that since we use topology as the mathematical tool, in this section we focus on functions from continuous spaces to continuous spaces.

In Section 4, we discuss the practical implications of surjectivity of these building blocks, using concrete examples in language, vision, and robotics. In particular, we show that Transformers, diffusion models, and certain policy networks commonly used in robotics are all surjective. Our work highlights significant obstacles to achieving provably safe architectures in these applications. In particular, applications on language modeling involves dealing with discrete tokens. In this part, we also discuss how we could extend our results into approximate surjectivity in discrete spaces in this section.

**Broader Implications of Surjectivity on Safety.** In Section 4.4, we discuss the implications of surjectivity on model safety and safety training. Existing work on jailbreaks has made important progress by identifying and mitigating specific vulnerabilities. However, without a deeper understanding of whether such vulnerabilities are avoidable in principle, research on jailbreaks runs the risk of becoming a cat-and-mouse game of patching symptoms rather than addressing root causes. Our work complements these efforts by offering a more foundational perspective on jailbreaks that highlights a fundamental challenge in creating jailbreak-proof safe AI models, using surjectivity as the formalism.

From a theoretical perspective, surjectivity implies that a model is vulnerable to jailbreaks in principle. That is, every outcome, including those considered harmful by model providers, can be generated by some input. The study of surjectivity also neatly decouples risks that are rooted in an attacker's ability to elicit particular behavior — which is the main consideration of jailbreaks — from the domain-specific risks that arise from having highly capable AI models in certain areas (such as bioweapons, etc.) in the first place. Given that our results hold under no particular assumptions on the training process (other than acknowledging that elements in the optimization pipeline introduce randomness in the training process), this shows that, at least in theory, safety training on several commonly used model architectures cannot prevent the model from outputting harmful behavior. Still, surjectivity is an existential property that does not guarantee that inputs for eliciting harmful behavior can be found with efficient computation or a feasible amount of information. We discuss these considerations further in Section 4.4, highlighting how some existing attacks can be viewed through the lens of surjectivity, what the study of surjectivity adds to the discourse on complementary approaches to safety training for AI safety, and exploring future directions for work that might be of interest to the community.

More broadly, the surjectivity of modern architectures prompts a deeper question about the state of research in AI safety: what are the appropriate frameworks for studying safety, jailbreaking, and even copyright risks in generative models? In all three cases, current evaluations often rely on probing the model's output behavior to examine whether certain harmful behavior or output resembling proprietary information can be elicited through adversarial inputs. But given that surjectivity implies that any output can, in principle, be elicited from any model, our work suggests that caution is needed when drawing conclusions about model safety based solely on its output behavior — especially when inputs can be manipulated outside typical usage patterns.

## 2    NOTATIONS AND PRELIMINARY

We denote individual vectors by letters $x, y$, and sequence vectors using $a, b, c$. When using subscript, $x_i, y_i$ mean the $i$-th entries, while $a_i, b_i, c_i$ mean their $i$-th elements. Symbol $\| \cdot \|$ represents the 2-norm of a vector or matrix. Symbol $\odot$ indicates entry-wise multiplication between vectors or matrices. Symbol $\oplus$ represents the direct sum (Cartesian Product) of linear spaces and we use it to define input spaces for sequences. For a positive integer $n \in \mathbb{N}^+$, we define $[n] = \{1, \cdots, n\}$. The identity matrix of dimension $d$ is represented by $I^d$. For a set $\Omega \subset \mathbb{R}^d$, Let $\partial\Omega$ be its boundary and $\overline{\Omega}$ be its closure. For a function $f : \mathbb{R}^d \to \mathbb{R}^d$ and a set $S \subset \mathbb{R}^d$, we denote $f(S) = \{f(x) | x \in S\}$.

### 2.1    NEURAL NETWORKS

Next, we give an overview of common modern neural network building blocks. Let the input of the network be a vector $x \in \mathbb{R}^d$ where $d \in \mathbb{N}$ is the input dimension. The output $y \in \mathbb{R}^d$ is also a vector. In this paper, we only discuss networks with the same input and output dimensions. One of the most elementary architectures is the Multi-Layer Perceptron.

**Definition 1.** *An $m$-layer Multi-Layer Perceptron (MLP) is a function $f : \mathbb{R}^d \to \mathbb{R}^d$ defined as*

$$f(x) = \sigma_m(W_m \cdots \sigma_2(W_2\sigma_1(W_1x + \lambda_1) + \lambda_2) \cdots + \lambda_m)$$

*where $\{W_i\}_{i\in[m]}$ are trainable matrices, $\{\lambda_i\}_{i\in[m]}$ are trainable vectors called* bias terms*, and $\{\sigma_i\}_{i\in[m]}$ are nonlinear entry-wise functions called* activation functions*. Row dimensions of $W_1, \cdots, W_{m-1}$ are called* hidden dimensions*.*

We define common examples of activation functions, namely ReLU, Leaky ReLU, and GeLU (Hendrycks & Gimpel, 2023) below:

$$\text{ReLU}(x)_i = \max\{x_i, 0\}, \text{LeakyReLU}(x)_i = \max\{x_i, \alpha x_i\}, \text{GeLU}(x)_i = x \cdot \frac{1}{2}\left[1 + \text{erf}\left(\frac{x_i}{\sqrt{2}}\right)\right].$$

Here the subscript $i$ means the $i$-th entry of a vector, $\alpha \in (0,1)$ is a preset constant and $\text{erf}(\cdot)$ is the Gauss error function.

Residual connection (He et al., 2016) and layer normalization (Ba et al., 2016) are essential elements of modern deep neural networks that usually work together.

**Definition 2.** Layer Normalization[1] *is a function* $\text{LN} : \mathbb{R}^d \to \mathbb{R}^d$ *defined as*

$$\text{LN}(x) = \gamma \odot \frac{x - \overline{x}}{\|x - \overline{x}\|/\sqrt{d}} + \beta, \quad \overline{x} = \frac{1}{d}\sum_{i=1}^{d} x_i$$

*where $\gamma, \beta \in \mathbb{R}^d$ are trainable parameters. When used with residual connections, there are two variants called* Pre-LayerNorm *and* Post-LayerNorm*. When wrapped around a function $f : \mathbb{R}^d \to \mathbb{R}^d$, residual connection, Pre-LayerNorm and Post-LayerNorm are respectively defined as*

$$g(x) = f(x) + x, \quad g(x) = f(\text{LN}(x)) + x, \quad g(x) = \text{LN}(f(x) + x).$$

Pre-LayerNorm has become common practice in modern neural networks as it stabilizes training (Xiong et al., 2020; OpenAI et al., 2024; Touvron et al., 2023; Chowdhery et al., 2023). Post-LayerNorm is also used sometimes (Zhuo et al., 2025; Li et al., 2024a; Vaswani et al., 2017).

---

[1]In real-world implementation a small $\varepsilon$ is added to the denominator in LN. Here we omit it due for presentation simplicity and it does not change the proofs in this paper.

**Sequence Models.** There are also specialized architectures dealing with sequential data, which take in an input sequence $a_1, \cdots, a_n$ and output another sequence $b_1, \cdots, b_n$. Attention (Bahdanau et al., 2016; Vaswani et al., 2017) is one of the most widely used sequence models.

**Definition 3.** *For trainable parameters key, query and value matrices $K, Q, V \in \mathbb{R}^{d \times d}$ A causally-masked attention layer calculates outputs as*[2]

$$b_i = \text{Attn}(a)_i = \frac{1}{Z_i} \sum_{j=1}^{i} \exp\left(a_j^\top K^\top Q a_i\right) V a_j, \quad Z_i = \sum_{j=1}^{i} \exp\left(a_j^\top K^\top Q a_i\right).$$

Causally-masked attention layers are used for autoregressive generations. Specifically, given input $a_1, \cdots, a_n$, the next token is generated from decoding $b_n$. After that this token is appended to the input and subsequent tokens are iteratively generated in the same way. There are also variants to $\text{Attn}$ called linear attentions (Yang et al., 2024). The simplest linear attention is RetNet (Sun et al., 2023).

**Definition 4.** *For trainable parameters $K, Q, V \in \mathbb{R}^{d \times d}$, a Retention layer calculates output as*

$$b_i = \text{Ret}(a)_i = \sum_{j=1}^{i} \left(a_j^\top K^\top Q a_i\right) V a_j = S_i Q a_i, \text{ where } S_i = S_{i-1} + V a_i a_i^\top K^\top. \tag{1}$$

In other words, $\text{Ret}$ can be thought of as $\text{Attn}$ without the non-linearity introduced by the soft-max function though $\exp$ and $Z_i$. It admits a recurrent form as shown in Equation (1), so autoregressive generation becomes faster. Other variants keep the recurrent form and use more complicated update rules for better performance. To keep the presentation clean, we defer introduction of other variants to the Appendix. We also defer the discussions on multi-head attention to the Appendix.

**Transformer.** The Transformer architecture is widely used in many applications recently. Most of them can be expressed by the building blocks stated previously. To illustrate this, we take GPT-3 (Brown et al. (2020) referred to as GPT below) as an example here. A single block in GPT can be expressed as

$$b_i = \text{TF}(a)_i = W_2 \text{GeLU}(W_1 \text{LN}(c_i) + \lambda_1) + \lambda_2 + c_i, \text{ where } c_i = \text{Attn}(\text{LN}(a))_i + a_i.$$

Matrices $W_1, W_2^\top \in \mathbb{R}^{d \times d'}$. Besides, LN applying to a sequence means applying separately to each input vector, i.e. the layer norm of the $i$-th vector is $\text{LN}(a)_i = \text{LN}(a_i)$. In plain text, TF is an $\text{Attn}$ followed by a two-layer MLP, wrapped with Pre-LayerNorm respectively, and GPT is compositions of several TFs.

## 2.2 Differential Topology

Differential topology studies the properties of smooth manifolds that are invariant under smooth transformations. Since almost all neural networks are trained using back-propagation, the architectures are usually smooth.[3] Differential topology hence provides natural tools for us to prove surjectivity of neural networks. In this section, we go over the necessary mathematical concepts and results that will be used later. We restrict our scope to smooth maps $f, g : \mathbb{R}^d \to \mathbb{R}^d$ here unless otherwise specified and defer a more general introduction of differential topology to the Appendix.

One of the early triumphs of topology is the Brouwer's fixed point theorem.

**Theorem 2.1** (Brouwer's Fixed Point Theorem, (Dinca & Mawhin, 2021, Corollary 1.4.1))**.** *Let $B^d(R) = \left\{x \in \mathbb{R}^d \big| \|x\| \leq R\right\}$ be a $d$-dimensional ball with radius $R$. For every continuous function $f : B^d(R) \to B^d(R)$, there exists $x \in B^d(R)$ such that $f(x) = x$.*

This theorem can be generalized from a ball $B^d(R)$ to any convex closed bounded set. Most celebrated and common applications of Brouwer's fixed point theorem are in game theory, Economics, and the study of equilibria of dynamical systems.

---

[2]In practice, a scaling factor is inside the exp function (Vaswani et al., 2017), which we omit, as it does not affect our analysis.

[3]Some building blocks, such as ReLU are not smooth everywhere. However they are only not smooth on zero measure sets, otherwise we cannot obtain gradient for a substantial amount of inputs. Topology can still tackle this scenario because such functions can always be approximated by smooth functions. We refer interested readers to Hirsch (1976) and omit this subtlety for simplicity.

**Definition 5.** *Differential $Df : \mathbb{R}^d \to \mathbb{R}^{d \times d}$ is defined as $Df(x)_{ij} = \partial f_i(x)/\partial x_j$.*

Since $f(x)$ can be approximated linearly in a small neighborhood around $x$, $Df(x)$ describes the local behavior of $f$ around $x$. More specifically $Df(x)_{ij}$ describes the rate the $i$-th dimension of output changes with regard to the $j$-th dimension of input. Hence an invertible $Df(x)$ (equivalently one with $\det Df(x) \neq 0$) indicates that $f$ behaves well around $x$ in the sense that no small neighborhood containing $x$ is collapsed to be lower dimensional after the application of function $f$.

One of the key concepts in algebraic topology is homotopy. Roughly speaking, two continuous functions are homotopic if one can be continuously deformed to the other.

**Definition 6.** *Homotopy is a function class $\left\{ f_t : \mathbb{R}^d \to \mathbb{R}^d \middle| t \in [0,1] \right\}$ such that the associated $F : \mathbb{R}^d \times [0,1] \to \mathbb{R}^d$, defined by $F(x,t) = f_t(x)$, is continuous. For two functions $f, g : \mathbb{R}^d \to \mathbb{R}^d$, they are* homotopic *if there exists a homotopy such that $f_0 = f, f_1 = g$.*

Intutively, a homotopy is a function that connects two functions $f$ and $g$ in a smooth way. This concept feels abstract, but a concrete example will be given in the proof of Theorem 3.3. Now we are ready to introduce Brouwer degree, a generalization of the idea presented in Theorem 2.1. Degree theory is another powerful tool to prove surjectivity.

**Definition 7.** *(Dinca & Mawhin, 2021, Definition 1.2.4) Let $\Omega \subset \mathbb{R}^d$ be an open bounded set and $\overline{\Omega}$ be its closure. Let $\overline{f} : \overline{\Omega} \to \mathbb{R}^d$ be the restriction of $f$ on $\overline{\Omega}$. Then for any value $y \notin f(\partial \Omega)$ (i.e, any $y$ to which no $x$ on the boundary of $\Omega$ maps), the Brouwer degree is defined by*

$$\deg(f, \Omega, y) = \sum_{x \in \overline{f}^{-1}(y)} \operatorname{sgn} \det(Df(x)). \tag{2}$$

**Lemma 1.** *(Dinca & Mawhin, 2021, Theorem 1.2.2) If $f, g$ are homotopic, $\Omega$ is an open bounded set, and $v \notin F(\partial\Omega, t)$ for all $t \in [0,1]$, we have $\deg(f, \Omega, v) = \deg(g, \Omega, v)$.*

This property shows that Brouwer degree is homotopy invariant. This allows us to reduce the problem of calculating the degree of a complex function to that of a simpler one. Since nonzero degree implies there at least exists one term in the right hand side of Equation (2), it also implies existence of pre-image, which help us prove surjectivity of complex functions.

# 3 SURJECTIVITY OF ARCHITECTURAL BLOCKS IN MODERN NETWORKS

In this section, we analyze the surjectivity of fundamental building blocks in modern neural networks using tools from differential topology. Before diving into the details, we formalize the setting. Though elementary, we start with the formal definition of surjectivity.

**Definition 8.** *A function $f : \mathcal{X} \to \mathcal{Y}$ is* surjective*, if for any $y \in \mathcal{Y}$, there exists a pre-image, i.e., an $x \in \mathcal{X}$ such that $f(x) = y$. When $\mathcal{Y}$ is a subset of $\mathbb{R}^d$ with Lebesgue measure, function $f$ is an* almost surjective *function if for any $y$, except for a zero measure subset of $\mathcal{Y}$, nonempty pre-image exists.*

Surjectivity is closed under composition. Namely, if $f : \mathcal{X} \to \mathcal{Y}, g : \mathcal{Y} \to \mathcal{Z}$ are surjective, their composition $f \circ g$ is also surjective. This allows us to separately prove surjectivity of building blocks of neural networks and conclude that the whole network is surjective. It is often too good to hope that an architecture is always surjective with any parameter. For example, if we set $V = 0$ in attention (Definition 3), this layer will output nothing but zero. A less extreme example is when $V$ is not an invertible matrix, it is not possible to output a vector outside the subspace spanned by $V$'s column space. However, this almost never happens in practice because the set of non-invertible matrices takes up zero volume in the parameter space, and hence is almost never hit in trained models.

**Definition 9.** *Let $\mathcal{H}_\Theta = \{ f_\theta : \mathcal{X} \to \mathcal{Y} | \theta \in \Theta \}$ be a class of neural networks parameterized by $\theta \in \Theta$, where $\Theta$ is a subset of Euclidean space with Lebesgue measure. $\mathcal{H}_\Theta$ is an* almost always surjective *set, if $h_\theta \in \mathcal{H}_\Theta$ is an almost surjective function onto $\mathcal{Y}$, except for $\theta$ in a zero measure subset of $\Theta$.*

Below we analyze which architectures are almost always surjective and in Section 4 we discuss which real-world models are surjective when used in specific ways.

## 3.1 PRE-LAYERNORM

In this section, we prove that Pre-LayerNorm is surjective using a neat but nontrivial application of the Brouwer's fixed point theorem (Theorem 2.1)

**Theorem 3.1.** *Let $f : \mathbb{R}^d \to \mathbb{R}^d$ be a continuous function, then $g : \mathbb{R}^d \to \mathbb{R}^d$ defined by $g : x \mapsto f(\text{LN}(x)) + x$ is surjective*

*Proof.* By the definition of Pre-LayerNorm and triangle inequality, we know that $\|\text{LN}(x)\| < \sqrt{d}\|\gamma\| + \|\beta\|$. Since $f$ is continuous, we have $M = \sup_{x \in \mathbb{R}^d} \|f(\text{LN}(x))\| < \infty$. To prove surjectivity of $g$, we need to prove that for any $y \in \mathbb{R}^d$ there exists an input $x^*$ such that $g(x^*) = y$. Below we fix $y$ and prove such $x^*$ exists. Let $F : \mathbb{R}^d \to \mathbb{R}^d$ be a function defined by $F(x) = y - f(\text{LN}(x))$. Now we find a fixed point for $F$. Let $R = M + \|y\| + 1$, then by triangle inequality we have $\|F(x)\| < R$. Therefore, $F$ maps $B^d(R)$ into itself. We can thus define the restriction of $F$ on $B^d(R)$ as a function $F|_{B^d(R)} : B^d(R) \to B^d(R)$. By Theorem 2.1, we know that there exists $x^* \in B^d(R)$ such that $F|_{B^d(R)}(x^*) = x^*$. Plugging this $x^*$ back to the definition of $F$ we know $F(x^*) = x^* = y - f(\text{LN}(x^*))$. Thus $g(x^*) = f(\text{LN}(x^*)) + x^* = y$. This establishes that for any $y \in \mathbb{R}^d$ there exists a corresponding input $x^*$, so $g$ is surjective. $\square$

This is a powerful theorem in the sense that it places minimal requirements on the function $f$. Nearly all modern neural networks are continuous, so the theorem implies that any architecture wrapped with Pre-LayerNorm is surjective. Notably, the proof does not rely on the specific expression of LayerNorm, but only on the fact that it is continuous and bounded. Hence if one uses other types of normalization functions with this property instead, like RMSNorm (Zhang & Sennrich, 2019), GroupNorm (Wu & He, 2018) or DyT (Zhu et al., 2025), this theorem still holds. Finally, this theorem can be easily extended to sequence models as we show below.

**Theorem 3.2.** *Let $f : \oplus_{i \in [n]} \mathbb{R}^d \to \oplus_{i \in [n]} \mathbb{R}^d$ be a continuous function, then $g : \oplus_{i \in [n]} \mathbb{R}^d \to \oplus_{i \in [n]} \mathbb{R}^d$ defined by $g(a)_i = f(\text{LN}(a))_i + a_i$ is surjective.*

The proof of Theorem 3.2 is deferred to the Appendix D. We also discuss Post-LayerNorm in Appendix C for completeness.

## 3.2 MLP: A FIRST APPLICATION OF DEGREE THEORY

As a warmup, we discuss the surjectivity of MLP (Definition 1). First we show that MLP with LeakyReLU is almost always surjective as long as the hidden dimension is at least the input dimension.

**Theorem 3.3.** *Two-layer MLP $f(x) = W_2\text{LeakyReLU}(W_1 x + \lambda_1) + \lambda_2$, where $x \in \mathbb{R}^d, \lambda_1 \in \mathbb{R}^{d_1}, \lambda_2 \in \mathbb{R}^d, W_1 \in \mathbb{R}^{d_1 \times d}$ ,and $W_2 \in \mathbb{R}^{d \times d_1}$ is almost always surjective when $d \leq d_1$.*

At a high level, this claim should come across as intuitive. Note that a full rank $W_1$ transforms the input space to a $d$-dimensional subspace of $\mathbb{R}^d$. LeakyReLU "bends" the subspace according to coordinate signs but does not change the way in which this subspace extends. Finally a full rank $W_2$ projects this bended subspace back to the original space $\mathbb{R}^d$, and is likely surjective. While intuitive, formalizing this intuition via linear algebra quickly becomes burdensome. One may try proving it by inverting the output by applying the inverse to $W_2$, LeakyReLU and $W_1$ consecutively. However when $d_1 > d$, $W_2$ is not invertible. Instead, we prove this using *homotopy* and demonstrate the power of differential topology in analyzing neural networks.

*Proof.* We start by constructing a simple surjective function $f^*$ and calculate its Brouwer degree, as defined on an appropriately defined open set (Definition 7). We then use Lemma 1 to connect $f$ and $f^*$ by a homotopy. This shows that $f$ has the same degree as $f^*$. The value of this degree will then be used to establish that $f$ is also surjective.

Let $f^*$ be a the same two-layer MLP as $f$ but with identity activation $f^*(x) = W_2(W_1 x + \lambda_1) + \lambda_2$. For any fixed value $v \in \mathbb{R}^d$, there is a unique solution to $f^*(x) = v$ which is $x^* = (W_2 W_1)^{-1}(v - \lambda_2 - W_2\lambda_1)$. Note that such a solution almost always exists, since $W_2 W_1$ is almost always invertible. Also note that invertible matrices have non-zero determinant. Therefore, for any

open bounded set $\Omega$ such that $x^* \notin \partial\Omega$, we have $\deg(f^*, \Omega, v) = \text{sgn} \det (W_2 W_1) \neq 0$. Next, we construct homotopy

$$F(x, t) = W_2 \sigma_t(W_1 x + \lambda_1) + \lambda_2, \text{ where } \sigma_t(x) = \max\{x, (t\alpha + 1 - t)x\}$$

such that $F(x, 0) = f^*(x)$ and $F(x, 1) = f(x)$. Intuitively, this homotopy is continuously changing $\alpha$ to 1 in LeakyReLU. Now comes perhaps the most abstract part of the argument. We need to construct $\Omega$ such that $v \notin F(\partial\Omega, t)$ for any $t \in [0, 1]$. We do this by showing that there exists some radius $R > 0$, such that $\Omega = \{x \mid \|x\| < R\}$ satisfies this property. In particular, irrespective of $t$, $\|F(x, t)\| > \|v\|$ for all $\|x\| = R$.

To see why, note that

$$\|F(x, t)\| \geq \|f(x)\| \geq \alpha\|W_2 (W_1 x + \lambda_1) + \lambda_2\| \geq \alpha (\sigma_{\min} (W_2 W_1) \|x\| - \|W_2 \lambda_1 + \lambda_2\|)$$

where $\sigma_{\min} (W_2 W_1)$ indicates the minimum singular value of matrix $W_2 W_1$. This implies that for

$$\|x\| = R = \frac{v/\alpha + \|W_2 \lambda_1 + \lambda_2\|}{\sigma_{\min} (W_2 W_1)}$$

we have that $\|F(x, t)\| > \|v\|$, establishing that $\Omega$ meets the conditions of Lemma 1.

Finally, by applying Lemma 1, we have that $\deg(f, \Omega, v) = \deg(f^*, \Omega, v) = 1$. Non-zero degree implies that $f^{-1}(v)$ is non-empty. Since this argument takes any values $v$ and shows that there is a pre-image for $v$, we have proved that $f$ is surjective. □

It is natural to ask whether MLPs with other activation functions are surjective. In contrast to LeakyReLU, many commonly used activations are not. Consider ReLU as an illustrative example and let us offer some intuition, while the formal statment and proof can be found at Theorem D.1. ReLU projects the subspace spanned by $W_1 x + \lambda_1$ to a manifold where all entries are positive. Hence there are many directions that $\text{ReLU}(W_1 x + \lambda_1)$ cannot reach. Let $v \in \mathbb{R}^{d_1}$ be one such direction. Then for substantially large norm $\|v\|$, the vectors near $W_2 v$ would never be reached by $f(x)$. Similar limitations apply to other activation functions with a constant lower bound, such as GeLU.

### 3.3 LINEAR ATTENTION

In recent years, linear attention mechanisms have gained significant attention due to their improved scalability compared to the original attention mechanism. Using degree theory introduced before, we are also able to analyze surjectivity for linear attentions. For example, for the Retention layer (Definition 4), we can prove the following theorem.

**Theorem 3.4.** Ret *is almost always surjective. However* Attn *is NOT almost always surjective.*

We defer the proof of the theorem and additional results to on linear attentions to the Appendix D. Note that the reason Attn is not almsot always surjective, while TF is surjective comes from the fact that the Attn and MLP in TF has Pre-LayerNorm, as mentioned in Section 3.1

## 4 IMPLICATIONS OF SURJECTIVITY ON MODERN APPLICATIONS

In this section, we discuss how our theoretical results on the surjectivity of large models from Section 3 relate to safety concerns in real-world settings, particularly with respect to adversarial attacks. Our goal is not to provide an exhaustive list of all practical models or attacks. Instead, we present concrete examples of generative models used across diverse application areas — such as language, vision, and robotics — to illustrate the scope and implications of the surjectivity theory.

### 4.1 LANGUAGE MODELS

As we established in Theorem 3.1 and Theorem 3.2, the Pre-LayerNorm used in every layer of all modern Transformer implementations is surjective. This shows that the modern Transformer architecture itself is also surjective.

**Corollary 4.1.** TF *is almost always surjective, and compositions of* TF *is almost always surjective.*

Our surjectivity results are proved in the embedding space under the assumption that we have direct control over the input embedding. This differs from the typical LLM deployments in two ways: First, language models operate on discrete tokens, which are mapped to embeddings via a fixed embedding function and arbitrary modification of token embeddings are not allowed. Second, most generative language models are decoder-only and operate autoregressively, generating one token at a time conditioned on previous outputs. Therefore, our theorems do not directly apply settings such as prompt-based attacks that exploit autoregressive generation (Zou et al., 2023). Arguably, this limitation might be reassuring for autoregressive models — had surjectivity held in that setting, it could imply a broader vulnerability surface for language models.

Despite the limitations, our results raise important questions about how one should interpret the input-output behavior of language models. For instance, concerns about copyright violations often cite the model's ability to reproduce specific outputs, such as a sentence from a proprietary source (He et al., 2025) as evidence. However, our findings imply that, in principle, any sentence can be produced by decoding the final layer output from a suitable input embedding, even if the model was trained for autoregressive generation. In the Appendix, we demonstrate this on GPT-2 (Radford et al., 2018), where we find a prompt such that decoding the last layer yields a 38-word sentence from a 2025 New York Times article, which could not have appeared in the model's training set. This suggests that caution is needed when inferring the presence of private or copyrighted data based solely on a model's output, especially when inputs are manipulated outside typical usage patterns. More broadly, this highlights the need for better metrics and processes for analyzing model safety from input-output behavior of models.

## 4.2 Vision Models

Diffusion models are a class of generative models originally proposed for image generation (Sohl-Dickstein et al., 2015). They are now widely applied to other domains, including video generation (Liu et al., 2024d), robotics (Chi et al., 2023), and beyond. Here we describe the generation process of diffusion models. Let the data we want to generate be represented by a vector $x \in \mathbb{R}^d$. First, one generates $x(0) \sim \mathcal{N}\left(0, I^d\right)$ from a Gaussian distribution. We regard $x$ as a variable depending on $t \in [0, 1]$. After that, $x$ evolves, or diffuses, according to a velocity vector field $v(x, t) \in \mathbb{R}^d$. The field $v$ is trained in the hope that $x(1)$ follows the same distribution as the data distribution $p(x)$ that we want to generate. Formally we have $\mathrm{d}x/\mathrm{d}t = v(x, t), x(0) \sim \mathcal{N}\left(0, I^d\right), x(1) \sim p(x)$. In practice directly solving this equation is often intractable, so we discretize $[0, 1]$ into intervals $\left\{[z_k, z_{k+1}]\right\}_{k \in [m]}$ with $z_1 = 0, z_m = 1$ and generate via approximation

$$x\left(z_{k+1}\right) = x(z_k) + v(x(z_k), z_k)(z_{k+1} - z_k), \text{for all } k \in [m]. \tag{3}$$

Early approaches to diffusion models inject Gaussian noise at inference time (see e.g. Ho et al., 2020), resulting in a stochastic updates that led to high quality outputs but slowing down the generation process. Here, we focus on the more recent *deterministic* ODE solvers (Song et al., 2022) as described above, which also benefit from faster generation. Velocity predictor $v$ is usually parameterized as a U-Net (Ronneberger et al., 2015; Ho et al., 2020) or Transformer (Peebles & Xie, 2023). When $v$ is implemented using a transformer, $x$ is tokenized into a sequence of vectors before being passed to the Transformer.

When using a diffusion model for image generation, a noisy image is first sampled as $x(0)$, and then diffuses according to $v$, resulting in a noiseless image $x(1)$. Recall Equation (3), since the first layer of $v$ is usually a normalization (usually GroupNorm for U-Nets and Pre Layernorm for Transformers), we can directly apply Theorem 3.1 or Theorem 3.2 to conclude that diffusion models are almost always surjective from the noise space to the output space. This structural property suggests an inherent vulnerability to adversarial attacks. Indeed, Zeng et al. (2024) constructed examples of $x(0)$s from the noise space, so that harmful contents are generated after the diffusion. Our results show that no matter how the diffusion model is trained, such input always exists for any output image.

## 4.3 Robotics

Neural networks, in particular sequence models, are increasingly common in robotics, and is making robots increasingly powerful. However, this also gives rise to safety concerns given our results. As

an example, let us take the policy network of Radosavovic et al. (2024), which follows a widely used design in practice. The network is implemented by a causally masked Transformer, i.e. compositions of TF. At timestep $t$, the action $b_t$[4] is generated by the Transformer on input sequence $a_1, b_1, \cdots, a_{t-1}, b_{t-1}, a_t$. Here $a$ is the sequence of observations from the environment.

Note that this sequence model diverges from the ones we studied in the earlier section, by interleaving the true input sequence (observations) and previous outputs (actions). This interleaving is done to improve the smoothness of robots actions. Using similar techniques from Section 3, we can prove that this policy network is almost always surjective.

**Theorem 4.2.** *Let* Rob *as compositions of* TF. *Given sequence $a$, we iteratively calculate sequence $b$ as $b_t = \text{Rob}(a_1, b_1, \cdots, b_{t-1}, a_t), t \geq 2; b_1 = \text{Rob}(a_1)$. This defines a function $f$ from $a$ to $b$. $f$ is almost always surjective.*

We defer the proof to the Appendix. This theorem means that for the policy network described in Radosavovic et al. (2024), almost any action sequence can be induced by some corresponding sequence of inputs — such as a video clip played for the robot — regardless of how undesirable or unsafe the resulting behavior may be.

### 4.4 Broader Discussion on the Implications of Surjectivity.

In this paper, we introduced the study of surjectivity of neural networks as a concrete formalization of studying the power of safety training and jailbreak vulnerabilities. Here, we will have a more detailed discussion. Due to space constraint, some of our discussion is deferred to Appendix F.

**On computational and statistical considerations.** From a theoretical perspective, surjectivity implies that a model is vulnerable to jailbreaks in principle. That is, every outcome including harmful ones, can be generated by some input. In practice, the relationship between surjectivity and jailbreak vulnerabilities is more nuanced. Surjectivity is an existential rather than a constructive statement: the existence of a pre-image $x$ for a harmful output $y$ does not imply that such a pre-image can be found computationally or information theoretically efficiently.

The proliferation of jailbreaks in practice suggests that computational difficulty is likely not a major bottleneck in the average case. While finding pre-images may be intractable in the worst case, worst-case hardness results do not provide meaningful safety guarantees, since attackers need only succeed on some harmful instances and may have significant computational resources.

From an information-theoretic perspective, surjectivity-based risks depend on the attacker's knowledge of the target outcome $y$. For example, in language domain, a model's surjectivity demonstrates moderate risk, as it could point to *repeated-after-me* attacks. Though this attack would let the model output harmful contents, it requires that the attacker knows the harmful text they want to elicit in detail. They cannot elicit sensitive hidden information that is a priori unavailable to the attacker (e.g., personally identifiable information, bio weapons risks). In other domains, such as robotics, the risk is more severe. For example, when outcome $y$ represents an action in the physical domain — such as the trajectory of an autonomous drone or a robot arm — the knowledge of the exact trajectory a drone must take to hit an object at a destructive speed may be known to the attacker. In these settings, surjectivity presents a significant concern as an attacker with knowledge of $y$ can craft an input that elicits the destructive behavior.

Still, many jailbreaks that do uncover new information can be framed in terms of surjectivity. For example, *suffix-injection* attacks (Wei et al., 2023; Zou et al., 2023; Wang et al., 2024c) work by eliciting outputs of the form $y = ab$, where $a$ is a fixed prefix (e.g., "Certainly! Here is ...", "Sure, the answer is ...") and $b$ contains novel potentially harmful information the model providers have sought to prohibit. Here the attacker's goal is not to learn anything from $a$ itself, but to ensure that the model commits to $a$ as the starting condition, exploiting the autoregressive continuation which make generating $b$ more likely. In other words, even though $a$ carries no new information, the ability to force such a prefix (as captured through the lens of surjectivity) significantly increases the probability of eliciting harmful $b$. More broadly, we believe that a natural direction for future work is to examine how much partial information about $y$ is sufficient to reconstruct a problematic input $x$ — for instance,

---

[4]Robotics convention uses $a$ for action, but we use $b$ here to keep notations in this paper consistent.

quantifying what fraction of the output tokens must be fixed or how many adaptive queries an attacker can make before finding an $x$ that produces an output in the vicinity of $y$.

**On Implications of Surjectivity on Safety Interventions.** Surjectivity also bears on the discourse in AI Safety community, in particular about two types of overall practical approaches to safety, which we summarize as "train-for-safety" (Grattafiori et al., 2024; Ko et al., 2024) versus "filter-for-safety" (Inan et al., 2023; Shi et al., 2025). The former includes works on post-training for safety, RLHF/RLAIF, and safety pre-training that aim to bake restrictions on the output space into model weights at training time, while that latter includes task-specific filters and constitutional classifiers that post-hoc filter the generated outcomes. The train-for-safety methods offer low-latency and efficient generation policies that are highly desirable to the model providers compared to post-hoc filtering that discard harmful generated response. The surjectivity of a model can be taken as further evidence that train-for-safety paradigm is not a sufficient line of defense on its own.

**On Surjectivity versus Model Capability.** Model capability perhaps is best captured through details of the input-output behavior. By design, our study of surjectivity is agnostic to whether the input-output relationship a model captures is a complex or an interesting one. Indeed, even simple functions, such as the *identity* function, can be surjective, while they are often not capturing interesting input-output behaviors. Our study of surjectivity intentionally decouples risks that are rooted in an attacker's ability to elicit particular behaviors — which is the main consideration of jailbreaks — from domain-specific risks that arise from having highly capable AI models in certain areas (such as bioweapons, etc.) in the first place.

One can take for granted that models that have undergone safety-training (and as result are the main subject of the study of jailbreaks) are already highly-capable models that have been trained on troves of relevant data to capture the complex relationship between the input and output spaces that deviates significantly from simple functions such as the identity. By establishing that these trained models are almost always surjective, our work highlights the inherent vulnerability of them, regardless of how capable the models are. That is, with enough information and computational power, an attacker can elicit any behavior from the model, including harmful ones.

**On Surjectivity versus Impact of Implicit Regularization.** There is a line of work (Du et al., 2018; Arora et al., 2019) suggesting that standard optimization algorithms such as gradient descent provides implicit regularization, encouraging the models to bias towards lower-rank parameters. A natural question is whether such structure would impact our analysis on being almost always surjective. We would like to remark that the "low-rank" phenomena typically correspond to approximately low-rank matrices whose singular values decay but are not exactly zero in practice. These matrices are still full rank, and hence the corresponding linear maps have range equal to the full output space. From the perspective of our results, such models remain in the generic region where surjectivity holds; the main effect of regularization is to make certain output directions correspond to small singular directions, so that reaching them may require large-norm or highly atypical inputs. This is precisely the regime in which adversarial optimization can search for off-distribution inputs that exploit these directions, which is consistent with the "in-principle" vulnerability captured by surjectivity.

**On embedding versus token spaces.** Our work studies surjectivity in the embedding space. Mathematically, our toolbox of differential topology requires continuous domains. Conceptually, we also see the study of surjectivity in the embedding space as increasingly relevant — e.g., due to the rise in multi-modal generative models that take continuous inputs (such as image patches). We also find that the assumption aligns with some white-box jailbreak methods that have access to the embedding space and optimize over it.

Additionally, one may hope to be able to extend our work to embeddings induced by discrete tokens. Informally, one could hope to obtain an approximate notion of surjectivity at the token level, under additional assumptions on the tokenizer and model properties. For example, if the token embedding matrix produces a sufficiently dense set of vectors in the embedding space (this might happen when embedding dimension is relatively small compared to the size of the vocabulary) and the network has some lipschitzness in its inputs, then for any hypothetical embedding sequence $x$ there is a real token sequence $\tilde{x}$ for which $f(\tilde{x}) \approx f(x)$. This would yield an approximate notion of surjectivity at the token level. In the current work, we do not attempt to formalize such conditions, but we believe this is an interesting direction for future work.

## 5 REPRODUCIBILITY STATEMENT

We state the assumptions in the main text and include proofs for our theoretical results in the Appendix. We also include the experiment details in Section E.

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

# A    RELATED WORKS

**Invertible Architectures.**    Invertibility, the property of being both injective and surjective, is a stronger notion than surjectivity which has been studied in neural networks before. Rezende & Mohamed (2015) proposed normalizing flow, which uses invertible functions to model complex distributions. A line of work constructing invertible neural networks thus follows (Dinh et al., 2015; 2017; Kingma et al., 2016; Papamakarios et al., 2017; Kingma & Dhariwal, 2018; Durkan et al., 2019; Chen et al., 2019). Beyond density estimation, researchers also explored invertible networks motivated by memory savings and representational power (Gomez et al., 2017; Jacobsen et al., 2018; Behrmann et al., 2019; Song et al., 2019). Specifically, in sequence models invertible architectures have been proposed to save memory (MacKay et al., 2018; Kitaev et al., 2020; Mangalam et al., 2022). A key difference to our work is that while prior efforts aimed to modify architectures to ensure invertibility, the modern architectural blocks we study were not designed with invertibility in mind.

**Safety.**    Attacks on generative models that lead to safety violations have been extensively studied in prior works.

We start with Language Models. There is a long line of work studying jailbreaks, which means constructing prompt to elicit undesirable behaviors from a trained language model. Jailbreaks can be classified as black-box attacks and white-box attacks. Black-box attacks restrict the attacker's access such that only prompt inputs are allowed, and we do not have knowledge about the model's internal parameters or architecture. We list some methods as follows. Goal-hijacking guide the model to override intention of the original prompt, and follow the attacker's wish by adding additional prompt to the original prompt (Perez & Ribeiro, 2022; Liu et al., 2024b). Another similar method suppresses the model from refusing to answer harmful questions (Wei et al., 2023). Few-shot jailbreaks manipulate the model by showing it demonstrations of harmful responses (Rao et al., 2024; Wei et al., 2024; Li et al., 2023). Code jailbreaks take advantage of model's power of code comprehension to conceal malicious contents in codes (Kang et al., 2024; Liu et al., 2024a). A line of work tells the language model to role play in a fictional world to let it generate harmful outputs (Liu et al., 2024c; Deshpande et al., 2023; Shah et al., 2023; Kang et al., 2024; Xu et al., 2024). Some attacks exploit lack of alignment data in low-resource languages to achieve jailbreak (Yong et al., 2024; Deng et al., 2024; Xu et al., 2024). A similar approach jailbreaks models by communicating in a ciphered texts (Wei et al., 2023; Yuan et al., 2024; Handa et al., 2025). In contrast to black-box approaches, white box approaches allow us to access the whole model (open-source models). The seminal work of Greedy Coordinate Gradient (Zou et al., 2023) use gradient-based method to optimize a suffix in the embedding space to maximize the likelihood of harmful output. Subsequent works following this path and improve the success rate by better optimization strategies(Zhang & Wei, 2025; Hu et al., 2024; Jia et al., 2024; Liu et al., 2024a; Huang et al., 2024; Geisler et al., 2025; Huang et al., 2025; Sitawarin et al., 2024; Wang et al., 2025a). Notably, some works attacks the model through manipulating hidden embedding (Wang et al., 2024c; Hu & Wang, 2024). Allowing access to model weights also allow us to finetune the model from safe to unsafe ones Wan et al. (2023); Rando & Tramèr (2024); Liao & Sun (2024); Kumar et al. (2024); Paulus et al. (2024). Other safety concerns of generative language models include bias, privacy, misuse, agent safety, and so on. We refer interested readers to Shi et al. (2024) for a more comprehensive survey.

Vision models has also become very powerful in recent years, and safety concerns rises. When saying vision models we usually refer to vision-language models (VLMs), because most useful vision models nowadays include both modalities. Let us first talk about white-box attacks. One line of works exploits the vision module by constructing adversarial images to let the model output undesirable images or texts Qi et al. (2024); Schlarmann & Hein (2023); Bailey et al. (2024); Madry et al. (2019); Luo et al. (2024); Shumailov et al. (2021); Chen et al. (2022). Another line of works attacks the model by exploiting both modalities (Wang et al., 2024a; Li et al., 2024b; Luo et al., 2024; Ying et al., 2024). For VLMs, there is a special class of attacks called grey-box attacks. These methods leverage the fact that a lot of vision encoders are CLIP (Radford et al., 2021) or BLIP (Li et al., 2022) to create better attacks. Like white-box attacks, there are also single-modality (Zhao et al., 2023; Dong et al., 2023; Niu et al., 2024) and cross-modality (Shayegani et al., 2023) attack methods. Last we introduce black-box methods. One line of works attacks models by constructing malicious typography (Gong et al., 2025; Qraitem et al., 2025; Wang et al., 2024b; Teng et al., 2025). In these methods malicious information is embedded into pictures that could have been rejected through text input. Other attacks

include using visual role play (Ma et al., 2024b), exploiting visual understanding capabilities (Zou et al., 2024) and so on. We refer interested readers to a more comprehensive survey paper by Ye et al. (2025).

Attacks in robotics is not studied as extensively as language and vision generative models. Common attacks to visual inputs include gradient-based pixel-level attacks (Du et al., 2022; Goodfellow et al., 2015) and patch-based attacks that can be realized in physical world (Athalye et al., 2018; Xu et al., 2020). There are also recent works on attacking vision-language-action models (Wang et al., 2025b).

## B  LINEAR ATTENTION

In this section we introduce other variants of Linear Attentions besides $\mathrm{Ret}$. We do not intend to give a comprehensive survey about all variants here. Instead we give a general introduction and some examples, and discuss how the proof of surjectivity for $\mathrm{Ret}$ can be extended in the next section. For a more comprehensive summary we refer readers to Yang et al. (2024; 2025). As stated in Yang et al. (2024), a lot of architectures can be written as

$$b_i = S_i Q a_i, \text{ where } S_i = G_i \odot S_{i-1} + V a_i a_i^\top K^\top.$$

Here $G_i \in \mathbb{R}^{d \times d}$ depends on $a_i$ and can be thought of as controlling which entries of $S_{i-1}$ should be retain and which should be forgot. For example, in Mamba-2 Dao & Gu (2024) we have

$$G_i = \gamma_i \mathbf{1}^\top \mathbf{1}, \text{ where } \gamma_i = \exp\left(-\mathrm{softplus}(\Gamma a_i)\exp(a)\right)$$

where $\Gamma \in \mathbb{R}^{1 \times d}$ us a trainable row vector and $a \in \mathbb{R}$ is a trainable parameter. $\mathbf{1}$ is a $d$-dimensional vector with every entry being 1. Mamba-2 introduce a decay on every entry of $S_i$ with the same rate depending on $a_i$. In contrast RWKV-6 (Peng et al., 2024) introduces a different decay to each column of $S_i$ be setting

$$G_i = \mathbf{1}\alpha_i^\top, \text{ where } \alpha_i = \exp\left(-\exp\left(A a_i\right)\right)$$

where $A \in \mathbb{R}^{d \times d}$ is a trainable matrix. There are also other variants that can not be expressed in this way. For example, in DeltaNet (Schlag et al., 2021), the update for $S_i$ can be expressed as

$$S_i = S_{i-1}\left(I^d - \beta_i K a_i a_i^\top K^\top\right) + \beta_i V a_i a_i^\top K^\top, \text{ where } \beta_i = \sigma(\Gamma a_i)$$

where $\Gamma \in \mathbb{R}^{d \times d}$ is a trainable matrix and $\sigma$ is an activation function mapping input to $[0, 1]$.

## C  POST LAYER NORMALIZATION

Here, we briefly discuss Post-LayerNorm for completeness, and as an additional example of how the geometric tools introduced in Section 2 can be useful. Before showing the results, we introduce the Inverse Function Theorem as follows.

**Theorem C.1** (Inverse Function Theorem). *Let $x \in \mathbb{R}^d$ satisfy $\det Df(x) \neq 0$, then there exist open sets $U \ni x, V \ni f(x)$, such that $f$ is bijective between $U$ and $V$.*

Since LN normalizes its input, we cannot expect a Post-LayerNorm network to reach every point in $\mathbb{R}^d$. More precisely, LN can only output values in set $S = \gamma \cdot \left\{x \in \mathbb{R}^d | \|x\| = 1, \overline{x} = 0\right\} + \beta$. Hence if we want Post-LayerNorm to be surjective on $S$, it suffices to show that $f(x) + x$ can reach any direction. A sufficient condition for this is that the image of $f(x) + x$ contains an open set containing 0, which can be easy to prove by Theorem C.1. As an instance, we can prove the following result.

**Theorem C.2.** *Let $f : \mathbb{R}^d \to \mathbb{R}^d$ be an MLP without bias term, with GeLU activation, and the hidden dimensions are all $d$, then $\mathrm{LN}(f(x) + x)$ is almost always surjective on $S$.*

*Proof.* Let $g : \mathbb{R}^d \to \mathbb{R}^d$ be $g(x) = f(x) + x$, and derivative $s = \mathrm{GeLU}'(0)$, then by the chain rule

$$Dg(0) = sI^d \cdot W_m \cdot sI^d \cdots sI^d \cdot W_1 + I = s^m \prod_{i=m}^{1} W_i + I$$

which is almost always full rank and therefore its determinant is non-zero. Since $g(0) = 0$, by Theorem C.1, we know that these is almost always an open set $V \ni 0$ such that $g$ is surjective on $V$. Hence for any vector $v \in \mathbb{R}^d$, there exists real positive number $\mu > 0$ such that $\mu v \in V$ because otherwise $0$ is on the boundary of $V$, contradicting the definition of open set. Moreover, since LN normalizes the inputs to unit vectors, we have $\text{LN}(v) = \text{LN}(\mu v)$, so $S = \text{LN}(\mathbb{R}^d) = \text{LN}(V)$. $\square$

*Remark* 1. We do not aim to provide a comprehensive result for MLPs with Post-LayerNorm. Many of the conditions in this proposition can be relaxed. The network architecture can be quite flexible, as long as its determinant almost never vanishes and the zero input maps to zero output. Moreover, MLPs without bias terms are not uncommon in practice (Groeneveld et al., 2024; Touvron et al., 2023; Chowdhery et al., 2023).

Notice that surjectivity on $S$ cannot be applied directly to compositions of functions, because this would change the domain of the next function to $S$ too. We leave the study of surjectivity of Post-LayerNorms, from $S$ to $S$, for future work.

## D  OMITTED PROOF IN SECTION 3 AND SECTION 4

**Theorem 3.2.** *Let* $f : \oplus_{i\in[n]}\mathbb{R}^d \to \oplus_{i\in[n]}\mathbb{R}^d$ *be a continuous function, then* $g : \oplus_{i\in[n]}\mathbb{R}^d \to \oplus_{i\in[n]}\mathbb{R}^d$ *defined by* $g(a)_i = f(\text{LN}(a))_i + a_i$ *is surjective.*

*Proof.* The proof is very similar to that of Theorem 3.1. Let

$$M = \sup_{a\in\oplus_{i\in[n]}\mathbb{R}^d} \|f(\text{LN}(a))\| \leq \infty.$$

For any specific output sequence $b \in \oplus_{i\in[n]}\mathbb{R}^d$ that we want to find corresponding input, construct $R = M + \|b\| + 1$. Then applying Theorem 2.1 on function $F(a) = b - f(\text{LN}(a))$ restricted to $B^{nd}(R)$ we prove the existence of corresponding input. Hence $g$ is surjective $\square$

Like Theorem 3.1, this proof can also be extended to other norms that are continuous and has bounded output.

**Theorem D.1.** *Two-layer MLP* $f(x) = W_2\text{ReLU}(W_1x + \lambda_1) + \lambda_2$, *where* $x \in \mathbb{R}^d, \lambda_1 \in \mathbb{R}^{d_1}, \lambda_2 \in \mathbb{R}^d, W_1 \in \mathbb{R}^{d_1 \times d}$ *and* $W_2 \in \mathbb{R}^{d \times d_1}$ *is not almost always surjective.*

*Proof.* We are going to prove that for $W_2$ from the following subset $\Omega$, $f$ is not surjective:

$$\Omega = \left\{ W_2 \in \mathbb{R}^{d \times d_1} | W_{1i} > 0 \text{ for all } i = 1, \cdots, d_1 \right\}$$

This set has positive measure in the parameter space of $W_2$, hence proving this statement would prove that the $f$ is not almost always surjective. By definition of ReLU, we know that $\text{ReLU}(W_1x + \lambda_1) \geq 0$, hence $[W_2\text{ReLU}(W_1x + \lambda_1)]_1 \geq 0$ when $W_2 \in \Omega$. Thus, $f(x)$ cannot reach any vector whose first entry is smaller than $\lambda_{21}$. In conclusion, $f$ is not almost always surjective. $\square$

In the following we separately prove the two statements in Theorem 3.4.

**Theorem D.2.** Ret *is almost always surjective.*

*Proof.* By definition, we need to prove that for any output sequence $b_1, \cdots, b_n$, there exists a corresponding input sequence $a_1, \cdots, a_n$. The proof is by induction on the output sequence. The first output $b_1$ only depends on $a_1$:

$$b_1 = a_1^\top K^\top Q a_1 V a_1 \Rightarrow a_1 = \left( \left(V^{-1}b_1\right)^\top K^\top Q V^{-1}b_1 \right)^{1/3} V^{-1}b_1.$$

This solution implicitly assumes that $V$ is invertible and $\left(V^{-1}b_1\right)^\top K^\top Q V^{-1}b_1 \neq 0$, both excluding zero measure sets from the Euclidean space. The induction hypothesis is: when searching for a pre-image of $b_j$ for $j > 1$, pre-images $a_1, \ldots, a_{j-1}$ are already determined and our choice of $a_j$ can only depend on $b_j$. That is, we want to solve for $a_j$ that meets the following requirement:

$$b_j = S_j V a_j + V a_j a_j^\top K^\top Q a_j,$$

where $S_j$ is the recurrence used in Definition 4 as a function of $a_1, \ldots, a_{j-1}$. Solving for such $a_j$ explicitly is difficult. Instead, we show that this map is surjective using our next Lemma, whose proof is based on the degree analysis.

**Lemma 2.** *(Informal) Function $f : \mathbb{R}^d \to \mathbb{R}^d$ defined by $f(x) = Mx + (x^\top Nx)x$ is almost always surjective with $M, N \in \mathbb{R}^{d \times d}$ as parameters.*

In the proof we construct homotopy $F(x, t) = tMx + (x^\top Nx)x$. We defer the detailed proof to Appendix. But let us show how such a lemma comes in by massaging the previous equation into a form that is more easily handled by the lemma. Since $V$ is invertible, let $a_j' = Va_j$ and define $b_j$ as function of $a_j'$:

$$b_j = f\left(a_j'\right) = S_j a_j' + a_j' a_j'^\top \left(V^{-1}\right)^\top K^\top Q V^{-1} a_j' = M a_j' + \left(a_j'^\top N a_j'\right) a_j',$$

where matrices $M = S_j, N = \left(V^{-1}\right)^\top K^\top Q V^{-1}$. Using Lemma 2 concludes the proof. $\qquad\square$

We put the proof of Lemma 2 because it is a lot more tricky than the others.

**Theorem D.3.** Attn *is not almost always surjective.*

*Proof.* We can first simplify the proof to surjectivity from $a_t$ to $b_t$ just like The proof of Theorem 3.4. If $K^\top Q$ is not semi-positive-definite, which is true with high probability if we choose parameters randomly when $d$ is big, there exist a lot of vectors such that $a_t^\top K^\top Q a_t < 0$. These vectors are either the volume of cone $a_t^\top K^\top Q a_t = 0$ or its complement. In these directions the norm of $\exp\left(a_t^\top K^\top Q a_t\right) Va/Z_t$ is upper bounded, and hence hence for output in such direction with sufficiently large norm, a corresponding input does not exist. In conclusion Attn is not almost always surjective. In fact with high probability it is not surjective if we choose parameters randomly according to some absolutely continuous distribution when $d \gg 1$. $\qquad\square$

**Theorem 4.2.** *Let* Rob *as compositions of* TF. *Given sequence $a$, we iteratively calculate sequence $b$ as $b_t = \text{Rob}(a_1, b_1, \cdots, b_{t-1}, a_t), t \geq 2; b_1 = \text{Rob}(a_1)$. This defines a function $f$ from $a$ to $b$. $f$ is almost always surjective.*

*Proof.* The proof is by induction which resembles the proof of Theorem 3.4. Output $b_1$ only depends on $a_1$. By Theorem 4.1 we know that the $b_1 = \text{Rob}(a_1) = \text{TF}(a_1)$ is surjective. When constructing $a_j$, we assume that all $a_1, \cdots, a_{j-1}$ has already be determined by $b_1, \cdots, b_{j-1}$. In this way, $b_j$ only depends on $a_j$, and the dependence is through a function which is a composition of functions with Pre-LayerNorms. By Theorem 3.1 we know that this function is surjective. In conclusion, Rob is surjective. $\qquad\square$

## D.1 Proof of Lemma 2

In this part we state the proof of Lemma 2 and discuss how the same proof strategy can be extended to other Linear Attention architectures.

Before presenting the proof let us first outline the proof idea. As stated in the main body, the proof is by constructing the following homotopy:

$$F(x, t) = tMx + \left(x^\top Nx\right)x, \ t \in [0, 1], f(x) = F(x, 1)$$

and we attempt to use that the degree of $v$ does not change as $t$ changes. The tricky part of this proof is that there exist directions $x \in \mathbb{R}^d$ such that $\left(x^\top Nx\right)x = 0$, so the construction of $\Omega$ is not straightforward. Put it in another way, there might exist roots to equation $F(x, t) = v$ going to infinity as $t \to 0$, making the construction of bounded set $\Omega$ such that no root crosses its boundary difficult. Notice that since such roots can only go to infinity along directions where $x^\top Nx = 0$, we may 'dig out' the small cone wrapping around $x^\top Nx = 0$ and replace it with a more well-behaved function, and consider the modified function instead. Here a well-behaved function should go to infinity even if $x$ goes to infinity along $x^\top Nx = 0$. However such modification is tricky for function

$F(x, 0) = (x^\top N x) x$, because when we cross $x^\top N x = 0$ the output vector turns to an almost opposite direction. To circumvent this, we instead consider function $g : \mathbb{R}^d \to \mathbb{R}^d$ defined as

$$g(x) = Mx + |x^\top N x| x$$

We will show that as long as this function is almost always surjective, we are able to show that $f$ is almost always surjective. The formal proof is as follows.

First let us define the modified function $h : \mathbb{R}^d \to \mathbb{R}^d$ as

$$h_\delta(x) = \begin{cases} |x^\top N x| x & \text{if } |x^\top N x| > \delta \|x\|^2 \\ \delta \|x\|^2 x & \text{if } |x^\top N x| \le \delta \|x\|^2 \end{cases}$$

where $\delta \in \mathbb{R}^+$ is a small positive number. This function 'digs' out the ill-behaved region between $x^\top N x = \pm \delta \|x\|^2$ and replace it with a function that goes to infinity whenever $\|x\| \to \infty$. Notice that $h$ is a continuous function.

**Lemma 3.** *For any $\delta \in \mathbb{R}^+$, equation $\hat{g}(x) = Mx + h_\delta(x) = v$ has solution for any $v \in \mathbb{R}^d$. Here $x \in \mathbb{R}^d$ is the variable, matrix $M, N \in \mathbb{R}^{d \times d}$ are fixed and $N \ne 0$.*

*Proof.* The proof is by constructing homotopy

$$\widehat{G}(x, t) = tMx + h_\delta(x), t \in [0, 1], \widehat{G}(x, 1) = \hat{g}(x).$$

Since $\|h_\delta(x)\| = o(\|x\|^2)$ and $\|Mx\| = O(x)$, we have $\widehat{G}(x, t) = o(\|x\|^2)$ for any $t \in [0, 1]$. Thus there exists a bounded set $\Omega$ such that $\widehat{G}(x, t) \ne v$ for any $x \in \partial\Omega, t \in [0, 1]$. $\widehat{G}(x, 0) = h_\delta(x) = v$ has exactly one solution for $v \ne 0$ with nonzero degree. Hence by Lemma 1 $\deg(\hat{g}, \Omega, v) \ne 0$, and hence $\hat{g}(x) = v$ has a solution. Besides, for $v = 0$, the equation has solution $x = 0$. In conclusion $\hat{g}(x) = v$ has solution for any $v \in \mathbb{R}^d$. $\qquad\square$

**Lemma 4.** *For almost all any $v \in \mathbb{R}^d$, there almost always exists a $\delta > 0$, such that there is no solution to $\hat{g}(x) = Mx + h_\delta(x) = v$ with $|x^\top N x| \le \delta \|x\|^2$. Here $M, N$ are matrices of some fixed nonzero rank $r_M, r_N$, which can be smaller than $d$.*

*Proof.* This statement should come across as intuitive because when $\delta$ is small, the ill-behaved region, namely the region where $|x^\top N x| \le \delta \|x\|^2$, is transformed to a small region around $\{Mx | x^\top N x = 0\}$, which is a manifold with less dimension than $d$. Here we give a proof that is rigorous in the mathematical analysis sense.

To prove the original statement, let us prove the contrapositive statement: If vector $v$ satisfies that for any $\delta > 0$ there exists an $x$ such that $\hat{g}(x) = v$ and $|x^\top N x| \le \delta \|x\|^2$, vector $v$ is constrained in a zero measure set.

From $|x^\top N x| \le \delta \|x\|^2$ for any $\delta > 0$ we know that the solution $x$ must satisfy $x^\top N x = 0$. Notice that $x^\top N x = 0$ is already a zero measure set for nonzero $N$, and for any specific $\delta$, set $\{\hat{g}(x) | x^\top N x = 0\}$ also has measure zero. Hence $v$ satisfying the conditions are constrained in a zero measure set. In conclusion, such $\delta$ almost always exists. $\qquad\square$

So far we have proved that the modified function has a root, and with appropriate $\delta$ the existence of root can be transferred to the original function. Now we state the complete version of the Lemma 2 as Lemma 5. In the main body we omitted the fact that $M, N$ are not freely chosen in the matrix space. Instead they are chosen from sets of matrices with fixed rank, depending on position in the sequence, which can be lower than full rank.

**Lemma 5.** *Function $f : \mathbb{R}^d \to \mathbb{R}^d$ defined by $f(x) = Mx + (x^\top N x) x$ is almost always surjective. Here $M, N$ are matrices of some fixed nonzero rank $r_M, r_N$, which can be smaller than $d$.*

*Proof.* By Lemmas 3 and 4 we know that $g(x) = Mx + |x^\top N x| x$ is almost always surjective. For some fixed $M, N$, let $P = \mathbb{R}^d \backslash \text{Im} g$ be the set of points that are not reachable by $g$. Let $g'(x) = -Mx + |x^\top N x| x$ and $Q = \mathbb{R}^d \backslash \text{Im} g'$. Equivalently, $-Q$ is the set of points that are not

reachable by $Mx - |x^\top Nx|x$. Let $\mu$ be the Lebesgue measure. Since $x^\top Nx$ either equals $|x^\top Nx|$ or $-|x^\top Nx|$

$$\mu\left(\mathbb{R}^d \backslash \mathrm{Im} f\right) \leq \mu(P) + \mu(-Q) = 0.$$

Hence the set of points not reachable by $f$ is zero measure. In conclusion, $f$ is almost always surjective. $\qquad\square$

The proof strategy presented here is applicable to a lot of variants of Linear Attention. The proof takes advantage of the fact that $\left(x^\top Nx\right) \to \infty$ for almost all $x$ except for those in cone $x^\top Nx = 0$. In order to make sure roots do not emerge from infinity along this cone as we vary $t$, we cut out the region near this cone, replace it with a well-behaved function. By the same argument, we can see that Mamba-2 (Dao & Gu, 2024) and RWKV-6 (Peng et al., 2024) are both almost surjective, since they do not change the fact that $f(x) \to \infty$ in most directions. We do not intend to prove surjectivity for all architectures as they are too many of them and they iterate fast. However we expect a similar proof strategy to work for other architectures like DeltaNet (Schlag et al., 2021), where the behavior of the function as $x \to \infty$ becomes more complicated. We leave the analysis of other architectures for future work.

## E  EXPERIMENT

Our results prove surjectivity results of practical architeuctres by proving surjectivity of their building blocks. Hence our proof also provides algorithms to find input corresponding to the a specific output of the network that we want. For GPT-2, or more generally surjective autoregressive models, we can find the input one by another as described in Algorithm 1.

---
**Algorithm 1** Finding Input Sequence
---
**Input:** A Frozen Transformer TF, An Output Sequence $b$
**Output:** A Reconstructed Sequence $a$
    $a_1 \leftarrow 0$
    Optimize $a_1$ using gradient descent on loss $(b_1 - \mathrm{TF}(a_1))^2$
    **for** $i = 2$ to $n$ **do**
        $a_i \leftarrow 0$
        Optimize $a_i$ using gradient descent on loss $(b_i - \mathrm{TF}(a_1, \cdots, a_i)_i)^2$
    **end for**
    **return** $a$
---

In this section, we conduct experiments to verify the theoretical statements using GPT-2. In particular we implement Algorithm 1 to find inputs corresponding to the following outputs:

- Twenty sentences from New York Times 2025. GPT-2 could not have been trained on such sentences. One example is 'The United States and China said Monday they reached an agreement ... threatening the world's two largest economies'. The length of these sentences varies from 7 words to 37 words.
- Twenty five sentences that contain completely random words from vocabulary, with length 2, 4, ..., 50. One example is "whims produ ether debunked depressive FoundingeeshedonApplication Weight refin 58".

Notice that Algorithm 1 cannot guarantee that we always find a corresponding $a$ because gradient-based optimization is still a heuristic algorithm. However, decoding one input by another is a lot simpler that jointly optimizing the whole sequence $a$ at a time. If this optimization is still too difficult, we can further decompose the algorithm into finding the hidden embeddings iteratively. However we find that for GPT-2 Algorithm 1 is enough.

For every gradient descent, we set learning rate to be 0.1 and optimize for 200 steps. We use an A100 GPU for inference. For all forty five sentences we described above, the algorithm succeeds in finding the corresponding input sequences. The decoding speed per token is $10.25 \pm 0.14$ seconds.

# F    ADDITION BROADER DISCUSSION ON THE IMPLICATIONS OF SURJECTIVITY

**On Theoretical Implications of Surjectivity.**    One major goal of safety training is to limit model's ability generate harmful outcomes. From a theoretical perspective, surjectivity implies that a model is vulnerable to jailbreaks in principle. That is, every outcome including those that are considered harmful by the model providers, can be generated by some input. We make no claim in the other direction. In particular, it is possible that many or even every harmful behavior by a non-surjective model can still be elicited by some input, while the model's lack of surjectivity is due to its inability to produce other non-harmful behavior.

**On Surjectivity Versus Having Full Support.**    One might ask how surjectivity is different from a common assumption that generative models have full support? This question stems from viewing the outcome of the neural network as a stochastic function from input to the output space, while in this work we view fully trained networks as deterministic function. When considering fully trained generative models as deterministic function, we find two perspectives to be instructive.

The first perspective is to consider generative models with deterministic decoding, e.g., decoding the probability distribution greedily, with beam search, or with temperatures close to $0$. In this perspective, the observation that a stochastic function has full support does not imply that their deterministic decoding schemes are surjective.

A second perspective is to consider the hidden embedding computed by the network as the deterministic function. Take for example GPT as a function that computes vector $b$ that is the hidden embedding output of its last block. Embedding $b$ is used by the model to output any token $i$ from a finite set of tokens with embeddings $h_i$, with probability $p(i \mid b) \propto \exp\left(b^\top h_i\right)$. GPT has full support, i.e., $p(i \mid b) > 0$ for all $i$, as long as $b$ is not parallel to any of the token embeddings. On the other hand, our surjectivity results allows one to deterministically generate any one of the tokens. Let's take an arbitrary token $i$. Surjectivity of the network implies that for some input, the transformer's hidden embedding is $b = \lambda h_i$ for large $\lambda$. Since $b$ is now parallel to the embedding of token $i$, $p(j|b) \to 0$ for all other tokens $h_j \neq h_i$. Thus surjectivity of the transformer implies that for any token $i$, there is some input that deterministically generates that token.

