# OpenReview forum: "On the Theory of Neural Network Surjectivity: Can You Elicit Any Behavior from Your Model?"
_ICLR.cc/2026/Conference — Submitted to ICLR 2026_

### Official Review · Reviewer_8qdq · 2025-10-15

**Soundness:** 2
**Presentation:** 3
**Contribution:** 3
**Rating:** 8
**Confidence:** 3

**Summary:**

The paper views generative models as functions and asks whether they are surjective - that is, whether every possible output can be realized by some input, with major implications on AI safety: it means that there exists some prompt for any possible model output, including harmful or jailbreaking ones.
Using tools from differential topology (homotopy, Brouwer degree), the authors prove that several modern architectures such as LeakyReLU MLPs, linear attention layers, and Pre-LayerNorm residual blocks—are almost always surjective.
They argue that this implies GPT-style Transformer blocks are surjective (in embedding space), raising concerns about the fundamental vulnerability surface of modern LLMs.

**Strengths:**

Framing surjectivity as a lens on model safety is novel and surprisingly elegant.
The use of degree theory and homotopy to prove surjectivity of nonlinear neural blocks is rigorous and instructive.
The connection between expressivity, topology, and AI safety is thought-provoking and could open new directions in theoretical AI safety research.

**Weaknesses:**

The discussion that the surjectivity theorems are proved only “in the embedding space under the assumption of direct control over the input embeddings” (p. 7) should be made much more prominent. To my understanding, this is a major conceptual limitation that belongs early in the Introduction, not as a short remark. The paper should clearly articulate the difference between embedding-level surjectivity and token-level generation in real autoregressive models. It would also be very helpful if the authors could comment on whether this assumption is merely an artifact of the mathematical analysis (which is my guess), or whether it (hopefully) represents a more fundamental limitation.

The triangle inequality seems to be used in the wrong direction in the derivation of R in Theorem 3.1. I believe a correct bound can be obtained by the other side of triangle inequality and by using the Lipschitz property of LeakyReLU and the minimal singular values of W1 and W2. Please verify this?

The link between the non-surjectivity of softmax attention and the surjectivity of GPT-style transformers could be explained more clearly. As I understand it, the Pre-LayerNorm residual connection makes any continuous sublayer surjective, which would reconcile why softmax attention alone is non-surjective but GPT (including softmax) blocks are. Could the authors make this reasoning explicit in the paper? I believe this is a CRUCIAL point.

“Smoothness” motivation is misleading: the claim that “since almost all networks are trained via backpropagation, they are usually smooth” (p. 4 line 190) is confusing, as in ML/optimization theory, “smooth” typically means Lipschitz gradient (C1 with bounded derivative). For the ICLR crowd, the standard notion of smooth function might not be the first thing coming to mind. ReLU networks are only piecewise-linear (C0, not C1). LeakyReLU is C1 but not C2. This is about presentation and not soundness, but I think is an important point.

The introduction of homotopy (Def. 6) feels abrupt. It should be motivated as the key technical tool allowing analyzing simple functions (linear) as a basis for more complicated ones (nonlinearities).
I suggest to also add after the definition a concrete example (for instance, the alpha-path from LeakyReLU to identity) to make the notion more tangible.

**Questions:**

You show that surjectivity holds in continuous embedding space, and even demonstrate it empirically by optimizing GPT-2 embeddings to reproduce arbitrary sentences.
However, could this theory be extended to embeddings induced by discrete token lookups?
For example:
Are there conditions on the embedding matrix or token distribution that preserve (approximate) surjectivity? Clarifying this would make the safety implications much stronger.

I will be happy to consider raising my score if these concerns are addressed, particularly if the embedding-space limitation is presented clearly in the Introduction and its possible relaxation is discussed.

**Details Of Ethics Concerns:**

The GPT-2 demo involves optimizing embeddings to reconstruct a publicly available sentence for illustration purposes. It does not constitute a harmful or privacy-related experiment.

---

> ### Author Response · Authors · 2025-11-26
> **Rebuttal**
>
> We thank the reviewer for their thoughtful comments and constructive feedback. The reviewer raised important questions regarding the generalization of our framework to discrete spaces. Below, we address these questions in detail and clarify the corresponding parts of the paper. We agree with the reviewer that including these remarks in the paper will improve the paper even more.
>
>
> > The discussion that the surjectivity theorems are proved only “in the embedding space under the assumption of direct control over the input embeddings” (p. 7) should be made much more prominent. [...] It would also be very helpful if the authors could comment on whether this assumption is merely an artifact of the mathematical analysis (which is my guess), or whether it (hopefully) represents a more fundamental limitation.
>
> > could this theory be extended to embeddings induced by discrete token lookups? Are there conditions on the embedding matrix or token distribution that preserve (approximate) surjectivity?
>
> We thank the reviewer for highlighting this important point and helpful suggestions. In the revised draft, we now state this restriction more prominently in the Introduction.
>
> Regarding whether this is merely an artifact of our analysis or a deeper limitation: It is a bit of both. Mathematically, our toolbox of differential topology requires continuous domains. Conceptually, we also see the study of surjectivity in the embedding space as increasingly relevant --- e.g., due to the rise in multi-modal generative models that take continuous inputs (such as image patches). We also find that the assumption aligns with some white-box jailbreak methods that have access to the embedding space and optimize over it.
>
> As for extending this theory to embeddings induced by discrete tokens, we find this a very interesting direction for future work. Informally, we believe that we can obtain an approximate notion of surjectivity at the token level, under additional assumptions on the tokenizer and model properties.
>
> For example, if the token embedding matrix produces a sufficiently dense and close set of vectors in the embedding space (this might happen when embedding dimension is relatively small compared to the size of the vocabulary) and the network has some lipschitzness in its inputs, then for any hypothetical embedding sequence $x$ there is a real token sequence $\tilde x$ for which $f(\tilde x)\approx f(x)$. This would yield an approximate notion of surjectivity at the token level. In the current work, we do not attempt to formalize such conditions, but we believe this is an interesting direction for future work. We have incorporated such a discussion into the Introduction and added a remark about this to section 4.4.
>
> > The triangle inequality seems to be used in the wrong direction.
>
> Thank you for pointing this out, we have fixed it in the updated version.
>
> > The link between the non-surjectivity of softmax attention and the surjectivity of GPT-style transformers could be explained more clearly.
>
> We have incorporated this suggestion in the updated version in Section 3.3.
>
> > “Smoothness” motivation is misleading
>
> We agree that some building blocks are not smooth everywhere, instead they are smooth everywhere except for a measure-zero set. Our intuitive explanation of smoothness here matches what backpropagation does in practice and essentially assumes that gradients are available for almost all points.
>
> Similarly, the toolbox of differential topology is amenable to working with functions that are smooth everywhere except for a measure-zero set of points. We omit this subtlety in this paper for simplicity. In the revised versions, we have incorporated these details in footnote 3.
>
> > The introduction of homotopy (Def. 6) feels abrupt.
>
> We add intuition to the concept and a pointer to the proof of Theorem 3.3 for a concrete example in the updated paper.
>
> We hope that our clarifications adequately address the reviewer’s concerns and help reassess the contribution in light of the updated explanations. If any part remains unclear, we would be glad to provide further details.

---

### Official Review · Reviewer_piGz · 2025-10-27

**Soundness:** 3
**Presentation:** 3
**Contribution:** 2
**Rating:** 2
**Confidence:** 3

**Summary:**

This paper studies whether different deep learning architectures are surjective by using tools from differentiable topology to prove this. The authors provide detailed theoretical results as well as an in depth discussion of the implications of these surjectivity results for AI safety. The authors argue that surjectivity poses a fundamental challenge to creating jailbreak-proof models since there will almost always exist an input that can lead a transformer model to produce any output.

**Strengths:**

- The paper is well written with clear explanations and intuitive introductions for all the relevant concepts.
- Theoretical results are clear and leverage interesting connections to differential topology.
- The proofs seem sound although some of them are outside of my area of expertise.
- The discussion is detailed and insightful

**Weaknesses:**

My key concern with this work is the following:

The authors claim that the surjectivity of certain DL models like transformers imply that they will always be susceptible to jailbreaks, however, as ankowledged by the authors, they do not account for discrete tokens or the autoregresive aspect of transformers. The latter seems particularly relevant since without autoregression, I struggle to think of outputs that are intrinsicly harmful and am therefore not convinced that surjectivity is a problem in itself.

If we are not looking at a non-autoregressive model like a classifier, then presumably we want the model to be able to predict any of the classes for some given inputs and the matter is more wether the some of the inputs cause the model to predict a class that does not match the human-assigned label (as in adversarial examples). In this case surjectivity is not the problem but the fact that the pre-image of a class contains samples that a human would assign to a different class. If we do not want the model to be able to predict a given class for any possible input, that class can be removed at an architectural level.

Similarly, for the drone case described in this work, any action in the action space of the AI controlled drone should be appropriate in some contexts. The problem would come from adversaries being able to induce this action in the wrong context, not if the adversaries create the right context for this action to be taken. Again I would say that surjectivity is not a problem in itself.

In the case of an autoregressive transformer based language model, for any token in the vocabulary there should be a sequence of inputs for which that token is the correct continuation. Again the problem comes from the kind of input that elicited the response and it is hard for me to imagine a single token response that could be intrinsically harmful. It is only when generating several tokens in a row or in relation to a specific prompt that I can think of intrinsically harmful or undesirable responses. For example if a language model responds "yes" to a user asking if they should do something illegal, that seems harmful. However, the fact that there exist prompts like "Say yes" for which the model will answer "yes" does not seem intrinsically problematic. Instructions on how to do something illegal seem like they would always be undesirable, no matter what they are in response to, however, as acknowledged by the authors, these multi-token generations where the adversary doesn't know the output they want to generate is not covered in this work.

**Questions:**

Am I missing something in the main weakness outlined above? Is surjectivity intrinsicly unsafe or are outputs only harmful in relation to a context?

What is prompt "a" found by your algorithm in Appendix D to elicit GPT2 to write a part of a New York Times article? Does the prompt include the target "b"?

Why do you claim that transformer blocks are surjective if you show that the attention mechanism is not surjective?

---

> ### Author Response · Authors · 2025-11-26
> **Rebuttal**
>
> We thank the reviewer for their review and feedback. The reviewer has several questions and comments regarding the implications of surjectivity, which we will answer below.
>
>
> Let us respond to a series of comments and questions as a whole, which we believe arise from viewing the outputs of neural networks as classification tasks rather than generative tasks. The focus of our paper is entirely on generative models and not on classification tasks. Let us elaborate:
>
> > [...]  I struggle to think of outputs that are intrinsically harmful and am therefore not convinced that surjectivity is a problem in itself.
>
> > If we are not looking at a non-autoregressive model like a classifier [...] If we do not want the model to be able to predict a given class for any possible input, that class can be removed at an architectural level.
>
> > [...] The problem would come from adversaries being able to induce this action in the wrong context, not if the adversaries create the right context for this action to be taken.
>
> >[..] we want the model to be able to predict any of the classes for some given inputs and the matter is more whether some of the inputs cause the model to predict a class that does not match the human-assigned label [to remedy this] the class can be removed at an architectural level.
>
> **Outputs being intrinsically harmful:** Generative models are capable of producing arbitrary outputs (rather than classifying inputs into one of many classes). These outputs can, in themselves, be harmful. As a concrete example, diffusion models for image or video generation could in principle produce child sexual abuse material, instructions for building a chemical weapon, or code implementing a computer virus. Such outputs are universally recognized as harmful by society and model providers. Regardless of the context, these outputs should not be produced by models available for public use under any circumstances.
>
> **On safety concerns in generative vs classification tasks:**
> For classifiers, we agree that surjectivity is usually not the central safety concern, and that certain classes can even be removed architecturally. By contrast, modern generative models (LLMs, diffusion models, robot policy networks) produce rich artifacts rather than a small discrete label set. In these settings, even describing all harmful outcomes is not feasible, and therefore removing them architecturally is not within reach.
>
>
> Because architectural removal of harmful outputs is not an option for generative models, model providers rely on safety-training --- i.e., training the network to avoid producing harmful outcomes $y$. Correspondingly, the study of jailbreaks for generative models focuses on whether attackers can circumvent this training and still produce some harmful outcomes $y$, by finding inputs $x$ such that $f(x) = y$. Our surjectivity results are precisely about this generative regime and provide a formal understanding of why such removal cannot be achieved at the level of training.
>
> **Answering other questions:**
>
> > as ankowledged by the authors, they do not account for discrete tokens or the autoregresive aspect of transformers.
>
> While our results do not apply to GPT-style decoder-only generative models at this moment, they have broader implications on many architectures used in practice, such as diffusion models and robotics, as discussed in Section 4. We agree that studying the autoregressive aspect of transformers is an interesting future direction.
>
>
> > What is prompt "a" found by your algorithm in Appendix D to elicit GPT2 to write a part of a New York Times article? Does the prompt include the target "b"?
>
> Since we operate at the embedding level, the input is not a sequence of language tokens. The prompt does not include the target. However, this does not mean that finding the prompt does not require knowledge of the target.
>
> > Why do you claim that transformer blocks are surjective if you show that the attention mechanism is not surjective?
>
> As defined in Section 2.1, the attention mechanism is wrapped around Pre-LayerNorm in Transformers. Hence, even if attention itself is not almost always surjective, by Theorem 3.2 we know that Transformers are surjective.

---

> > ### Comment · Reviewer_piGz · 2025-11-27
> > **Response**
> >
> > Thank you for your response. I think the image generation case you provided is a good example where an output can be intrinsically undesirable regardless of the context. However, I find examples like “code implementing a computer virus” less convincing since at least in the transformer case, this code would require multiple token generations and fall outside the scope of this paper.
> >
> > Both for robots and text-based transformers I remain unconvinced that a single forward pass can produce an output that is intrinsically undesirable. Each token or robot movement should be desirable in some context or architecturally removed.
> >
> > I would be open to increasing my score somewhat if the paper made the point that this has implications for image/video generation and maybe text based diffusion (not sure if the theory would hold in that setting) but limited its claims about its implications for text generation or robotics.
> >
> > On a related note, for the experiment in Appendix E, the algorithm only includes b1 which I take to be the first token of the target sequence. Does this mean you find a distinct sequence a to force each of the outputs in the target sequence? Overall I do not find Appendix E very clear.

---

> > > ### Author Response · Authors · 2025-12-03
> > >
> > > Thank you for your response. We are glad that you find it convincing that harm can inherently exist in the output itself.
> > >
> > > In a follow-up question, you asked whether the code generation example could, in principle, be outputted by a non-autoregressive model. The answer is yes. We note that non-autoregressive models do not lack expressivity compared to autoregressive models, and as such, these outcomes can be as easily generated by non-autoregressive models as they are by autoregressive models. Where the auto-regressive nature of a language model helps particularly is in hitting a good balance between expressivity and learning high-quality models within a reasonably sized training sample set. There is increasingly more evidence that the training of non-autoregressive language models (such as diffusion language models) can be done efficiently, which has led to high-quality and fast code generation ([1]). More fundamentally, while our paper does not study autoregressive Transformers, the formalism we provide encompasses autoregressive models, as the map from an input of the autoregressive model to the output is indeed a parameterized function that can be studied using our formalism. We do agree, though, that extending our results to autoregressive models more directly would be an interesting direction for future work.
> > >
> > > We believe that there might still be a misunderstanding about the robotics examples. In particular, let us clarify that the robotics example we provide requires more than one pass of the Transformer in its policy network. The policy takes in a stream of observation sequences and appends the output action to the end of the current observations, creating a variant of autoregressive generation.
> > >
> > > [1] Zhihui Xie, Jiacheng Ye, Lin Zheng, Jiahui Gao, Jingwei Dong, Zirui Wu, Xueliang Zhao, Shansan Gong, Xin Jiang, Zhenguo Li, and Lingpeng Kong. Dream-coder 7b: An open diffusion language model for code, 2025. URL https://arxiv.org/abs/2509.01142.

---

> > > ### Author Response · Authors · 2025-12-03
> > >
> > > Regarding Appendix E, we thank the reviewer for pointing out the typo here. In the decoding loop, we regress the output of the Transformer to $b_i$ instead of $b$. We have updated the PDF accordingly.

---

### Official Review · Reviewer_NQcw · 2025-10-29

**Soundness:** 1
**Presentation:** 2
**Contribution:** 1
**Rating:** 2
**Confidence:** 4

**Summary:**

The paper studies the surjectivity property of several models, including MLP with several activations, attention layers, and the effect of LayerNorm. The proof methods use tools from differential and algebraic topology, and show that several of the architectures are indeed surjective. An experiment is given in the appendix supporting the claim.

**Strengths:**

- The use of differential topology and degree theory is an interesting approach that I have not seen before in the literature of ML.

- The connection between surjectivity and safety is interesting, and could be further developed and also related to optimization rather than only expressiveness.

**Weaknesses:**

**W1 - Non-rigorous and hand-wavy proofs**

The paper contains many places with either non-rigorous proofs or hand-wavy arguments. This is a major problem, especially in a theoretical work. To give some examples:

- There is no formal proof for why MLP with ReLU is not surjective, and the informal intuition at the end of section 3.2  is very cryptic and hand-wavy. Table 1 shows the MLP with ReLU case as a result, which is clearly not the case here.

- In the proof of Theorem D.2, instead of providing a rigorous argument, it is said that the model “Cannot reach a large chunk of the output space “. This argument is very informal and not appropriate for a rigorous mathematical proof. To prove non-surjectivity formally, it is needed to show a specific output value that cannot be reached.

- Layer norm is defined incorrectly in this paper. The denominator should be the variance (plus a small $\epsilon$ term), so there is a $\frac{1}{\sqrt{d}}$ term missing. This also changes the proof of Theorem 3.1, and shows that the bound depends on the input dimension.

These are just some examples I found, I suggest going over the proofs and making sure every argument is justified.

**W2 - The motivation for studying surjectivity is lacking**

- The authors try to connect surjectivity to safety, which I think is interesting, but not properly aligned in the setting of this paper. For example, a linear predictor is also almost always surjective, so is it also not safe? The reason for surjectivity is the condition that $d_1 > d$, and so the linear part of the model is already surjective, and it is only left to show that the non-linearity doesn’t “ruin” this property, as in the case of ReLU. Since linear predictors are clearly not the focus of today’s safety issues, I believe the scope of the surjectivity property studies in this paper is very limited.

- Another point is that the condition of $d_1 > d$ is unrealistic in most of today’s models. For example, the input dimension of images is in the millions, while it is unrealistic to have a hidden dimension of the same size.

- The authors claim that the models are “almost always” surjective, excluding a zero measure set of low-rank matrices. However, recent results about the implicit bias of such models suggest that there is a tendency to converge to low-rank matrices, see e.g. Implicit Regularization in Deep Matrix Factorization, 2019, Arora et al., and follow-up works. This suggests that the zero-measure set is what models actually converge to; this should be at least mentioned in the paper.

**W3 - The experiment is lacking**

There is an experiment in the appendix that supports the surjectivity claim. However, it is very lacking and currently only hurts the surjectivity claim rather than supports it. First, the results are not clearly stated; it is said, “We run the algorithm many times for different sentences and it almost never fails. “ (line 1382). I don’t believe this is a good way to present the results in a scientific paper. Second, only a single sentence is presented, which is in standard natural language. An experiment should be done on several inputs, in a proper way, describing the results. Also, a more interesting result would be if the model could output non-realistic sentences or even gibberish, which should be guaranteed to be possible due to the surjectivity property.


A minor comment, I personally prefer putting the related works section in the main text and not in the appendix, as I view it as an integral part of the paper. It is better to move the proofs to the appendix if there are space issues.

**Questions:**

- Does surjectivity provide a practical mean to attack a model (and thus pose a safety issue), rather than a hypothetical possibility for an attack?

- In the more realistic setting where $d_1 << d$, is the model always not surjective? In this case, can you describe the space of outputs?

- Empirically, does the surjectivity property hold also for sentences not in natural languages? E.g., for random outputs.

---

> ### Author Response · Authors · 2025-11-26
> **Rebuttal (Part I)**
>
> We thank the reviewer for their review. The reviewer states three weaknesses to justify a score of 2. We respond to these weaknesses first, then respond to the reviewer's questions.
>
> **Weakness 1: Non-rigorous and hand-wavy proofs**
> We thank the reviewer for pointing out adjustments that can improve the paper and have reflected those in the revision. In particular, we have added a new theorem (Theorem D.1), removed an intuitive sentence in the proof of Theorem D.1, and added a missing $\sqrt{d}$ term in Definition 2. As we explain below, these are minor points that do not constitute a lack of rigor or handwavyness.
>
> > There is no formal proof for why MLP with ReLU is not surjective, and the informal intuition at the end of section 3.2 is very cryptic and hand-wavy.
>
> As our sentence ("let us offer some intuition without providing a formal proof") states, we intended to provide intuition here. In the updated version we add a formal proof in the appendix (Theorem D1) for interested readers.
>
> > In the proof of Theorem D.2, instead of providing a rigorous argument, it is said that the model "Cannot reach a large chunk of the output space."
>
> The sentence outlined by the reviewer is intended to provide intuition and the proof itself is rigorous without missing any details. In the updated version, we edit this sentence to "hence for output in such direction with sufficiently large norm, a corresponding input does not exist."
>
> > Layer norm is defined incorrectly in this paper.
>
> Thank you for pointing out a typo, we have corrected it. The correctness of Theorem 3.1 does not change, as it only relies on the output of LN being bounded.
>
> **Weakness 3: The experiment is lacking**
>
> Our paper is primarily a mathematically grounded theoretical study of surjectivity and was submitted to *learning theory (primary area)*, which focuses on theoretical contributions. We believe a theoretical study is most appropriate to provide a foundational perspective on jailbreak vulnerabilities that complements the extensive empirical literature happening in this space already. Such theoretical understanding will serve to ensure that research on jailbreaks transcends cat-and-mouse games of patching symptoms by better understanding root causes.
>
> That said, we included a small illustrative experiment in the supplementary material (Appendix E). The experiment was not to be an extensive empirical evidence of surjectivity (it is impossible to empirically validate surjectivity without attempting every possible output) rather to demonstrate one implication of surjectivity for copyright investigations: that it is feasible to numerically search for an input that produces a given output that was never part of the training dataset.
>
> > [experiments] currently only hurts the surjectivity claim rather than supports it.
>
> We understand how the original presentation may have led to this impression. As stated, the purpose of this experiment was to illustrate how the existence of a pre-image can be leveraged by simple optimization heuristics. In this sense, the experiments achieve exactly what is intended: it consistently finds an input embedding that maps to a specified output.
>
> We agree that expanded experiments are valuable in this case. In the revised version, we have expanded the experimental section to include a larger range of natural-language texts and randomly generated outputs. We have also released the code and data in the supplementary material for interested readers.

---

> > ### Author Response · Authors · 2025-11-26
> > **Rebuttal (Part II)**
> >
> > **Weakness 2: Motivation for studying Surjectivity**
> > Our paper includes significant discussion and motivation for studying surjectivity (e.g, more than a page in section 4.4 is dedicated to many facets of this discussion in addition to our introductory sections.) We will add a paragraph to Section 4.4 about the impact of implicit regularization to expand this discussion further and refer the reviewer to those paragraphs that address their comments:
> >
> > > For example, a linear predictor is also almost always surjective, so is it also not safe? Since linear predictors are clearly not the focus of today’s safety issues, I believe the scope of the surjectivity property studies in this paper is very limited.
> >
> > We refer the reviewer to the paragraph in section 4.4 titled "On Surjectivity versus Model Capability". Our framework is deliberately designed to decouple the existence of jailbreaks (formalized as: for a harmful outcome $y$, does there exist an input $x$ for which $f(x) = y$?) from the capabilities of the model, i.e., how rich or semantically meaningful the input–output relationship is. As we have stated, in this sense, we are fully in agreement that linear or identity functions can be surjective, while clearly not representative of modern highly capable models or the main focus of current safety concerns. Our goal is not to claim that surjectivity alone makes any surjective model “unsafe,” but rather that surjectivity makes a model vulnerable to jailbreaks.
> >
> > Indeed, one can take for granted that models that are the main subject of the study of jailbreaks and safety-training are already highly capable models that capture the complex input-output relationship that deviates significantly from the identity or linear functions. The main message of our paper is that no matter what other input-output properties are met by the trained neural networks (e.g., complex and highly non-linear functions versus simple identity of linear functions), those models are almost always surjective.
> >
> > To illustrate the implications of surjectivity, we gave examples from robotics, language models, and copyright. We do not presume that neural networks learned in these applications are linear or identity functions. But the fact that the learned neural network is almost always surjective constitutes a tangible risk, such as a drone taking a trajectory to hit a building at destructive speed.
> >
> > > Another point is that the condition of $d_1>d$ is unrealistic in most of today’s models. For example, the input dimension of images is in the millions, while it is unrealistic to have a hidden dimension of the same size.
> >
> > We agree that in some cases $d_1$ (hidden dimension) may be smaller than $d$ (output dimension), like in your example of using MLP to generate outputs with high-dimensional data. On the other hand, in many settings of practical relevance such as those included in Section 4.1 and 4.3, we are indeed in settings where $d_1=d$. Specifically, Transformers, as well as the robotics network from [1], uses MLPs with $d_1=d$, as discussed in Section 4.1 and 4.3. We will add a remark to clarify this further in the paper.
> >
> > >The authors claim that the models are “almost always” surjective, excluding a zero measure set of low-rank matrices. However, recent results about the implicit bias of such models suggest that there is a tendency to converge to low-rank matrices.
> >
> > We have added a paragraph to section 4.4 to address your remark. At a high level, while regularization could encourage the model to converge to low-rank matrices, these matrices are often only approximately low rank --- meaning their singular values decay quickly but are not necessarily zero. The distinction is that exactly low-rank matrices form measure zero subsets, while approximately low-rank matrices do not. When approximately low-rank models are exposed to adversarial attacks, the small but non-zero singular values of the matrices can be exploited to generate data in the full range of the output space. Even though, under standard non-adversarial inputs to the model, the low-rank structure might not produce the full range.

---

> > > ### Author Response · Authors · 2025-11-26
> > > **Rebuttal (Part III)**
> > >
> > > Next, we answer questions directly raised by the reviewer:
> > >
> > > > Does surjectivity provide a practical means to attack a model (and thus pose a safety issue), rather than a hypothetical possibility for an attack?
> > >
> > > The theory of surjectivity itself does not provide a practical means to attack. However, we have provided a heuristic algorithm and associated experiments (Appendix E) and discussion of attacks (Section 4.4 paragraph starting with "Still many jailbreaks") that show how such attacks can be constructed.
> > >
> > > > In the more realistic setting where $d_1<<d$, is the model always not surjective? In this case, can you describe the space of outputs?
> > >
> > > Please see our detailed response to Weakness 2 above. Briefly, MLPs with
> > > $d_1=d$ are used in most realistic applications, including in transformers and in robotic networks studied in Sections 4.1 and 4.3. When $d_1<d$, the model is never surjective, because the output is now a $d_1$-dimensional manifold, which is a strict subset of the $d$-dimensional Euclidean space.
> > >
> > > > Empirically, does the surjectivity property hold also for sentences not in natural languages? E.g., for random outputs.
> > >
> > > Surjectivity is the property of a function, not of its individual inputs or outputs. We take the reviewer's question as meaning whether the method from our Appendix E can work for finding inputs for a given output, where the output is not in natural language. The answer is yes, our method works for all given outputs in principle. To showcase this, we have expanded our empirical study and given additional experiments on finding inputs corresponding to output sequences that are randomly generated from the vocabulary of GPT-2.
> > >
> > > [1] Ilija Radosavovic, Tete Xiao, Bike Zhang, Trevor Darrell, Jitendra Malik, and Koushil Sreenath. Real-world humanoid locomotion with reinforcement learning. Science Robotics, 9(89):eadi9579, 2024.

---

### Official Review · Reviewer_cHvE · 2025-11-01

**Soundness:** 3
**Presentation:** 3
**Contribution:** 3
**Rating:** 6
**Confidence:** 3

**Summary:**

The paper examines the surjectivity of commonly used neural network layers, and networks composed of them - with potential implications for safety esp. in relation to jailbreaking and adversarial attacks. They do so by invoking various results from algebraic topology, e.g. Brouwer's fixed point theorem (Theorem 3.1) or using homotopy and Brouwer degree (Theorem 3.3). They classify various layer types as surjective or otherwise, and connect these results to commonly used architectures in ML. They then discuss the implications of their results for various domains and tasks.

I recommend the acceptance of the paper because it tackles an important, complementary aspect of safety discussion, and demonstrate the usefulness of a not commonly utilized mathematical toolkit in such analyses, potentially helping the field to advance. I do not recommend a strong acceptance because the paper itself does not make a strong connection between their results and practical safety concerns, nor does it suggest a convincing pathway to doing so.

**Strengths:**

- Examining the expressivity of the model architectures in question is an important and complementary aspect of ML safety. Although it certainly cannot be the only approach to safety, the authors' introduction sounds promising in terms of future research making more concrete connections between the surjectivity of function classes in question and achievability of certain adversarial outcomes.
- The analyses cover a good variety of modern neural network layers.
- The paper is written well and the main ideas are communicated clearly.
- The paper is clear about the limitations of its approach to safety, and extensively discusses how it relates to alternatives at a high level.

**Weaknesses:**

- Theoretical achievability of an outcome says little about its practical relevance. A dramatic example of this is adversarial attacks being by construction $\epsilon$-bounded around natural images. Although it is good that the authors are transparent about this fact, the fundamental problem still remains. The paper does not outline a convincing roadmap towards closing this gap.
- The paper's theoretical exposition also have some gaps in terms of the architectures they cover. For example, the MLP in Theorem 3.3 requires input and output dimensions to be equal, which is not unreasonable per se, but also leaves out important use cases for such architectures. It also ignores potential outcomes of optimization: e.g. explicit or implicit regularization not uncommonly leads to low-rank layer matrices.

**Questions:**

Here I add some more minor questions and comments:
- Page-long citations exist in the references, please fix.
- L034: Citation paragraph typo
- L125: That the current analyses address MLPs with $\mathbb{R}^d \to \mathbb{R}^d$ should be explicitly acknowledged and discussed, either here or later in the paper.
- L127: $b_2$ -> $\lambda_2$
- L196: Reference?
- L216: Would it make more sense to present Def. 7 and Lemma 1 closer to Theorem 3.3? I leave this up to the authors
- L216: Double parantheses. Also applies to Lemma 1.
- L283: What is this a warm-up to?
- L441: I had a hard time following the arguments in this paragraph; it would benefit from being rewritten more clearly.

---

> ### Author Response · Authors · 2025-11-26
> **Rebuttal**
>
> We thank the reviewer for their positive and careful review and valuable feedback. In the following we address the reviewer's questions:
>
> > Theoretical achievability of an outcome says little about its practical relevance. [...] The paper does not outline a convincing roadmap towards closing this gap.
>
> We would like to emphasize that the role of our theory is to elucidate the fundamental bottlenecks in AI safety theory. Without such understanding the discourse and research in AI safety runs the risk of becoming a cat-and-mouse game of patching symptoms rather than addressing root causes. One example of where we show an approach to closing this gap is in Section 4.4, where surjectivity bears on the discourse in AI safety community between "train-for-safety" and "filter-for-safety" showing that train-for-safety paradigm is not a sufficient line of defense on its own advocating for filtering methods as essential for closing the gap.
>
> > The paper's theoretical exposition also have some gaps in terms of the architectures they cover.
>
> We agree that our paper does not cover all practical architectures. Our goal has not been to give an exhaustive list of architectures, but rather to focus on the development of the framework of studying surjectivity of trained neural networks, analyzing many of the most commonly used architectures with concrete real-world use cases, as discussed in Section 4.
>
> > e.g. explicit or implicit regularization not uncommonly leads to low-rank layer matrices.
>
> We have added a paragraph to section 4.4 to address your remark. At a high level, while regularization could encourage the model to converge to low-rank matrices, these matrices are often only approximately low rank --- meaning their singular values decay quickly but are not necessarily zero. The distinction is that exactly low-rank matrices form measure zero subsets, while approximately low-rank matrices do not. When the model is exposed to adversarial attacks, the small but non-zero singular values of the matrices can be exploited to generate data in the full range of the output space. Even though, under standard non-adversarial inputs to the model the low-rank structure might not produce the full range.
>
> > Typos and writing suggestions
>
> We have fixed all the typos and incorporated the writing suggestions in the updated version.
>
> > What is this a warm-up to?
>
> We have changed this to state "First Application" in the updated version. We view Section 3.2 and Theorem 3.3 as being a more gentle and intuitive introduction to using the mathematical toolset compared to Theorem 3.4.
>
> > Line 196: Reference?
>
> We have added a reference to Brouwer's fixed point theorem.
>
> > L216: Would it make more sense to present Def. 7 and Lemma 1 closer to Theorem 3.3?
>
> Thank you for the suggestion. Our reasoning to state Definition 7 in Section 2 is to keep all the mathematical tools in the same section to allow a reader to more easily find them.
>
> > I had a hard time following the arguments in this paragraph; it would benefit from being rewritten more clearly.
>
> In the updated version, we rewrite this paragraph to improve clarity. In the current version, we start with pointing out that a repeat-after-me attack would let the surjectivity alone imply only moderate risk, and then argue that when the generative models induce physical consequences the risk is more severe.

---

### Author Response · Authors · 2025-12-03

We summarize the rebuttal at the beginning of the revised pdf.

---

> ### Author Response · Authors · 2025-12-04
> **For AC and SPC Consideration: Summary of the the rebuttal phase and the paper (part 1)**
>
> We appreciate your time and service as an Area Chair—particularly given the additional workload created by this year’s process changes. To make your assessment as easy and high-signal as possible, we begin with a concise summary of the paper, the reviews, and how the manuscript has improved through the rebuttal.
>
> In this paper, we investigate the topological properties of modern neural networks and show that many fundamental architectural components are almost always surjective. This property implies that when such architectures are used in generative models, the models are inherently vulnerable to jailbreak attacks: in principle, any output—including harmful ones—can be induced by a suitable input. Our contributions are strong on both the mathematical and conceptual sides: the topological analysis we develop not only establishes new and novel techniques for studying properties of trained networks on particular architectures (rather than just the level of expressivity of architectures), but also enables a principled understanding of why certain classes of generative models are vulnerable to jailbreaks.  **As reviewer 8qdq (score 8 and stating their interest in updating their input) puts it elegantly: “Framing surjectivity as a lens on model safety is novel and surprisingly elegant. The use of degree theory and homotopy to prove surjectivity of nonlinear neural blocks is rigorous and instructive. The connection between expressivity, topology, and AI safety is thought-provoking and could open new directions in theoretical AI safety research.”**
>
>
> --------
> We next address the two initially negative reviews. In one case, the concerns and scores arose from clear misunderstandings. In another, there are concerns with the review we highlight below. Based on the reviewers’ own comments during rebuttal, we believe that had they been able to continue engaging with the paper, both scores would have increased substantially.
>
> **Reviewer piGz (score 2)** raises concerns that stem from a misunderstanding of our problem setting—specifically, interpreting the outputs of neural networks through the lens of classification rather than generative modeling. Our paper focuses exclusively on generative models, where outputs are themselves content (images, text, code), not discrete labels. *This led to the reviewer’s central question: “can outputs be inherently harmful? Or, does harm arise from abusing the outcome of a model?”* Viewing outcome as possibly being harmful in themselves is a central feature of the research on AI Safety and prevention of jailbreaks.
>
> We addressed this clearly in our response, providing concrete and widely recognized examples of harmful content: a generative model producing child sexual abuse material; a language model generating instructions on synthesizing chemical weapons; or generating code that implements malware. In all these cases, the output itself is harmful regardless of downstream interpretation. As long as a model is surjective, such models are capable of producing such harmful contents given some inputs.
>
> **Importantly, Reviewer piGz explicitly stated that they found our examples convincing in demonstrating that harm can exist in the content itself. They also indicated willingness to revise their evaluation upward. We responded thoroughly to their other follow-up questions, e.g. on whether autoregression is needed for generating such outcomes. We believe their score would have increased substantially—likely to a strongly positive recommendation—had the rebuttal phase allowed them to update their rating.**

---

> > ### Author Response · Authors · 2025-12-04
> > **For AC and SPC Consideration: Summary of the the rebuttal phase and the paper (part 2)**
> >
> > **Reviewer NQcw (score 2)** provides an evaluation that diverges substantially from the rest of the reviewer panel in scoring and content. Their numeric ratings are outliers and are not supported by the content of their own review or the impression of other reviewers
> >
> > | |  Reviewer NQcw score | Avg Score of other reviewersColumn 3 |
> > |----------|----------|----------|
> > | Soundness   | 1  | 2.66  |
> > | Presentation | 2  | 3       |
> > | Contribution  | 1  | 2.66  |
> >
> > **Notably, the reviewer assigns a soundness score of 1 while not identifying any technical flaws or concrete issues in the paper. Their review does not contain any specific claims that could justify such an assessment, nor does it point to any errors (outside one simple typo) for us to address.** In addition, the reviewer did not participate in the rebuttal process before the reviews were frozen when we responded comprehensively to their concerns. Given the combination of (i) substantial deviation from other reviewers, and (ii) absence of technical justification for extremely low scores, we respectfully suggest that this review be given limited weight in the overall recommendation and the motive behind such extreme scoring be flagged.
> >
> > For completeness, we summarize how we have fully addressed the reviewer’s stated concerns:
> >
> > **W1: “Non-rigorous proofs.”**
> > We strongly disagree with this characterization. As researchers in the theoretical CS and learning theory communities, our standards for rigor are quite high and the proofs in this paper meet those standards. The reviewer’s comments focus on a few sentences they interpret as “hand-wavy,” but these sentences explicitly appear in intuition-building sentences where we deliberately opted for higher-level explanations. All proofs and formal arguments are done rigorously in the paper and the reviewer did not point out any concrete evidence of lack of rigor.
> >
> > **W2: “Insufficient motivation for studying surjectivity.”**
> > Our paper contains substantial discussion and motivation for surjectivity—including over a page in Section 4.4 alone—on top of the conceptual framing in the introduction. During the rebuttal, we added further clarifications on two specific points raised by the reviewer:
> >
> > Low-rank structure and surjectivity: As we explained, approximate low-rank structure arising from implicit or explicit regularization does not preclude surjectivity. This is because approximate low-rank (as opposed to exactly low rank) structures still produce output in the full span range of the output space.
> >
> > Relationship between input and output dimension: We addressed this concern in detail and highlighted a key fact: for the most common architectures considered in Sections 4.1 (transformers) and 4.3 (transformer-based robotic policy networks), the hidden dimension matches the output dimension exactly by definition. Thus, the parameter regime motivating our theory precisely corresponds to widely used practical architectures.
> >
> > Had the discussion phase been left open and the reviewer engaged during the rebuttal phase, we believe these clarifications would have fully resolved their concerns.
> >
> > **W3: “lack of experiments:** Our paper is primarily a mathematically grounded theoretical study of surjectivity and was submitted to the learning theory track (primary area: learning theory) reflecting that our primary contribution is theoretical. As Reviewer 8qdq notes, our framework is “novel and surprisingly elegant,” with a “rigorous and instructive” formulation whose connection between topology, expressivity, and AI safety “could open new directions in theoretical AI safety research.” Indeed, we believe that a theoretical study, rather then an empirical one, is the strength of our paper as it provides a foundational perspective on jailbreak vulnerabilities that complements the extensive empirical literature already happening in this space. Such theoretical understanding will serve to ensure that research on jailbreaks transcends cat-and-mouse games of patching symptoms by better understanding root causes.
> >
> > Nevertheless, we did include a small illustrative experiment in Appendix E. Its goal was not to empirically establish surjectivity. Indeed, it is impossible to establish surjectivity empirically without assessing every possible output of the model; this study has to be done theoretically.  But our experiments still established that simple heuristics can compute approximate preimages efficiently on arbitrary outputs of our choosing. During the rebuttal, we expanded this empirical section substantially with many more target outputs and corresponding recovered inputs, strengthening the illustrative component.

---

> > > ### Author Response · Authors · 2025-12-04
> > > **For AC and SPC Consideration: Summary of the the rebuttal phase and the paper (part 3)**
> > >
> > > Next, we want to highlight one of the positive and high quality reviews that we think deserved more attention. Overall, both of our positive reviews (scored 6 and 8) raised interesting questions that we fully answered in the rebuttal. We believe had the reviewers been allowed to continue engaging with the paper, they would have have raised their scores.
> > >
> > > **Reviewer 8qdq (score 8 and explicitly noting interest in raising it further) already viewed the paper as quite strong, stating: “Framing surjectivity as a lens on model safety is novel and surprisingly elegant. The use of degree theory and homotopy to prove surjectivity of nonlinear neural blocks is rigorous and instructive. The connection between expressivity, topology, and AI safety is thought-provoking and could open new directions in theoretical AI safety research.”**
> > >
> > > They also raised a deep and insightful question about how surjectivity in the embedding space might translate to approximate surjectivity in the token space. In response, we added extensive discussion to Section 4.4, which we summarize here: under additional assumptions on the tokenizer and model properties, one can obtain an approximate notion of surjectivity at the token level. For example, if the token embedding matrix produces a sufficiently dense and close set of vectors in the embedding space (this might happen when embedding dimension is relatively small compared to the size of the vocabulary) and the network has some lipschitzness in its inputs, then for any hypothetical embedding sequence $x$ there is a real token sequence $\tilde x$ for which $f(\tilde x)\approx f(x)$. This would yield an approximate notion of surjectivity at the token level.
> > >
> > > **As Reviewer 8qdq indicated, our clarifications and the additions to the manuscript would likely have led them to raise their score further.**

---

### Meta-Review · Area_Chair_HsJU · 2026-01-06

**Summary:**

In this paper, the authors propose an interesting perspective on model safety, which is to prove that generation models are surjective. Therefore, any output, including harmful or undesirable ones, can in principle be generated by the networks, leading to model safety and jailbreaks.

Strengths:

1. The proposed perspective is interesting.

2. The paper is well-written with clear theoretical results.

Weaknesses:

1. The findings are impractical. The surjectivity means the existence of x, but x can be impractical, like random tokens, which can be easily filtered by current prompt filters. And why safety training can mitigate jailbreaks is not deeply discussed in this paper.

2. Besides theoretical proof, the existence of surjectivity is not surprising. If you prompt an LLM or fine-tune an uncensored LLM to repeat any words of the inputs, they can generate anything you want.

In summary, although the proposed perspective is interesting, the paper's scenario is not practical and may not give much insight to society. Therefore, I suggest to reject.

**Reviewer Concerns:**

The reviewers' concerns mainly lie in the realism of his setup, the practical impacts of the results, and the actual experiments etc. I think they are not well-addressed.

**Reviewer Scores:**

I think the negative reviewers won't change their minds as their safety settings of safety is too simple and their practical insights are limited.

---

### Decision · Program_Chairs · 2026-01-26

Reject